# Precipitation thresholds regulate net carbon exchange at the continental scale

Zhihua Liu [1,2], Ashley P. Ballantyne [1], Benjamin Poulter[3], William R.L. Anderegg[4], Wei Li [5], Ana Bastos [5,6] & Philippe Ciais [5]

Understanding the sensitivity of ecosystem production and respiration to climate change is critical for predicting terrestrial carbon dynamics. Here we show that the primary control on the inter-annual variability of net ecosystem carbon exchange switches from production to respiration at a precipitation threshold between 750 and 950 mm yr$^{-1}$ in the contiguous United States. This precipitation threshold is evident across multiple datasets and scales of observation indicating that it is a robust result and provides a new scaling relationship between climate and carbon dynamics. However, this empirical precipitation threshold is not captured by dynamic global vegetation models, which tend to overestimate the sensitivity of production and underestimate the sensitivity of respiration to water availability in more mesic regions. Our results suggest that the short-term carbon balance of ecosystems may be more sensitive to respiration losses than previously thought and that model simulations may underestimate the positive carbon–climate feedbacks associated with respiration.

[1] Department of Ecosystem and Conservation Sciences, University of Montana, Missoula, MT 59812, USA. [2] CAS Key Laboratory of Forest Ecology and Management, Institute of Applied Ecology, Chinese Academy of Sciences, Shenyang 110016, China. [3] Biospheric Sciences Laboratory, NASA Goddard Space Flight Center, Greenbelt, MD 20770, USA. [4] Department of Biology, University of Utah, Salt Lake City, UT 84112, USA. [5] Laboratoire des Sciences du Climat et de l'Environnement/Institut Pierre Simon Laplace, Commissariat à l'Énergie Atomique et aux Énergies Alternatives–CNRS–Université de Versailles Saint-Quentin, Université Paris-Saclay, F-91191 Gif-sur-Yvette, France. [6] Department of Geography, Ludwig-Maximilians-Universität München, Luisenstr. 37, 80333 Munich, Germany. Correspondence and requests for materials should be addressed to Z.L. (email: liuzh811@126.com)

Terrestrial net ecosystem carbon exchange (NEE) currently absorbs the equivalent of approximately 25% of all anthropogenic $CO_2$ emissions[1] and plays a significant role in regulating the variability of the global carbon (C) cycle[2–4]. Despite the importance of terrestrial NEE, its response to climate is a major source of uncertainty in future climate predictions[5]. Terrestrial NEE represents the small imbalance between $CO_2$ assimilation through gross primary production (GPP) and $CO_2$ release through total ecosystem respiration (TER). GPP and TER are coupled over the long term through the distribution of carbon assimilated to ecosystem carbon pools and their subsequent turnover leading to TER. Yet, GPP and TER can be decoupled on temporal scales going from years to centuries if one of these fluxes is perturbed by environmental conditions, and small decoupled variations in GPP or TER fluxes can result in large variations in NEE. Studies have shown that inter-annual variability (IAV) of terrestrial NEE and its sensitivity to climate has increased during the past 50 years[3,6,7], but whether such an increase in NEE variability is due to the climate sensitivity of ecosystem production or respiration remains difficult to determine[8–11].

A rich history of multiyear site-based data has revealed that the sensitivity of ecosystem production to precipitation decreases as water availability becomes more abundant[12–17]. Global analyses suggest that fluctuations in global land NEE are either due to water-controlled production in dry land ecosystems[18,19], or due to temperature-controlled respiration in tropical ecosystems[3]. These competing hypotheses, i.e., water vs. temperature, were recently reconciled by Jung et al.[20] who found that water-driven GPP and TER responses compensate each other, dampening water-driven NEE variability regionally, and therefore leaves a dominant temperature signal at global scale. However, previous conclusions were either based on ecosystem-scale measurements, or based on the water availability and NEE simulated by global vegetation models, or on data-driven empirical models used to extrapolate NEE globally, which are difficult to verify at the regional scale. Therefore, we still lack a good empirical understanding of climate sensitivity of ecosystem production and respiration and their consequences on net terrestrial carbon dynamics at regional scales[21]. Using a dense network of well-constrained observations across the contiguous United States (CONUS), we began by testing whether the widespread assumption that production is the primary control on IAV of NEE is true at the continental scale. We then investigated what processes control the IAV of NEE and how do they respond to water availability. Finally, we studied whether the state-of-the-art dynamic global vegetation model (DGVM) can capture the production and respiration dynamics in response to continental-scale water availability patterns.

We first calculated the per-pixel temporal correlation between IAV of gridded observation-based fluxes (e.g., detrended GPP and NEE using GPP derived from MODIS observations of the fraction of light absorbed by plants using a light-use efficiency model ($GPP_{MODIS}$) and mean NEE from atmospheric $CO_2$ inversions constrained by a dense network of atmospheric $CO_2$ concentration observations, given atmospheric transport models ($NEE_{ACI}$) on an ecosystem scale using a dense network of 17 eddy covariance sites from across the representative biomes in CONUS (see Methods: Temporal correlation between IAV of GPP and NEE). Then, we compared the sensitivity of GPP and TER derived from these observation-constrained estimates with the results from an ensemble of ten DGVMs. Here precipitation was chosen as the main controlling variable for carbon fluxes, based on previous analyses[13–15] and our own sensitivity analysis (see Methods: Sensitivity analysis). Precipitation is a simple measure of ecosystem water availability that is accurately measured across CONUS, and there are several lines of evidence for strong

relationships with ecosystem production in this region[15]. Temporal sensitivities ($\delta^t$ and $\gamma^t$) were calculated from linear regression models in which ecosystem carbon fluxes (i.e., GPP or TER) are regressed against climate factors (i.e., precipitation and temperature) that varied over time. These values of $\delta^t$ and $\gamma^t$ indicate the apparent sensitivities of carbon flux anomalies to unit change in climate factor for a given ecosystem over time[14]. In addition, spatial sensitivities ($\delta^s$ and $\gamma^s$) were calculated from nonlinear models based on ecosystem flux data combined from grid cells or ecosystem sites and used to indicate the apparent sensitivities of mean carbon fluxes to a unit change in climate factor along climate gradients, generally across different ecosystem types[14]. Lastly, we conducted a simple respiration modeling experiment of increasing complexity based on empirically derived models of heterotrophic respiration (see Methods: Ecosystem respiration modeling experiments) in order to allow a process-oriented evaluation of DGVM results. We show a precipitation threshold between 750 and 950 mm yr$^{-1}$, below which the IAV of NEE is regulated by ecosystem production and above which IAV of NEE appears to be regulated by ecosystem respiration across CONUS. This precipitation threshold is evident across multiple datasets and scales of observation, but not captured by DGVMs, likely due to inaccurate simulation of heterotrophic respiration to environmental constraints.

## Results and Discussion

**Spatial and temporal correlation between GPP and NEE.** Contrary to the finding that ecosystem production is the primary factor controlling continental-scale variations in net carbon exchange[19,22,23], we find that mean annual GPP and NEE occupy different climate spaces (Supplementary Fig. 1) and do not necessarily covary spatially (Fig. 1a, b) or temporally (Fig. 1c) at regional scales. First, looking at spatial variations, ecosystem GPP is much more strongly controlled by mean annual precipitation (MAP; $r = 0.93$, $p < 0.001$) than by mean annual temperature (MAT; $r = 0.38$, $p < 0.001$), and increases with precipitation (Supplementary Fig. 2a), such that the highest mean annual GPP appears in the relatively warm and wet southeastern United States (Fig. 1a). In contrast, the largest NEE (i.e., a strong carbon sink) is found at intermediate levels of MAP (~750–1200 mm yr$^{-1}$), and then decreases at both higher MAP ( > 1200 mm yr$^{-1}$) and higher MAT (>20 °C) (Supplementary Fig. 2b), such that the highest mean annual NEE appears in the relatively cool and wet North Central United States (Fig. 1b). The spatial inconsistency between patterns of observed mean annual GPP and NEE suggests that the most productive ecosystems do not necessarily have the largest NEE uptake. This happens because NEE reflects not only current GPP but also the legacy of past ecosystem exposure to climate, the effect of management such as biomass harvest, and disturbances that decouple spatially annual NEE from GPP.

To test the commonly held assumption that photosynthesis is the main process regulating inter-annual NEE[24], we calculated Pearson's product moment temporal correlations between four independent GPP proxies and two NEE estimates at 1° spatial resolution from 2000 to 2014 across CONUS ($n = 8$ data products). Mean Pearson's $r$ from the resulting eight gridded observation-based fluxes showed that there is a significant positive correlation between GPP and NEE in more xeric (e.g., semiarid western grassland or shrubland) ecosystems, but only a very weak correlation in more mesic (e.g., eastern deciduous broadleaf forest) ecosystems (Fig. 1c). This sharply contrasting pattern is independent of the combination of GPP and NEE datasets used (Supplementary Fig. 3) and of their spatial or seasonal resolution (Supplementary Fig. 4), which suggests that

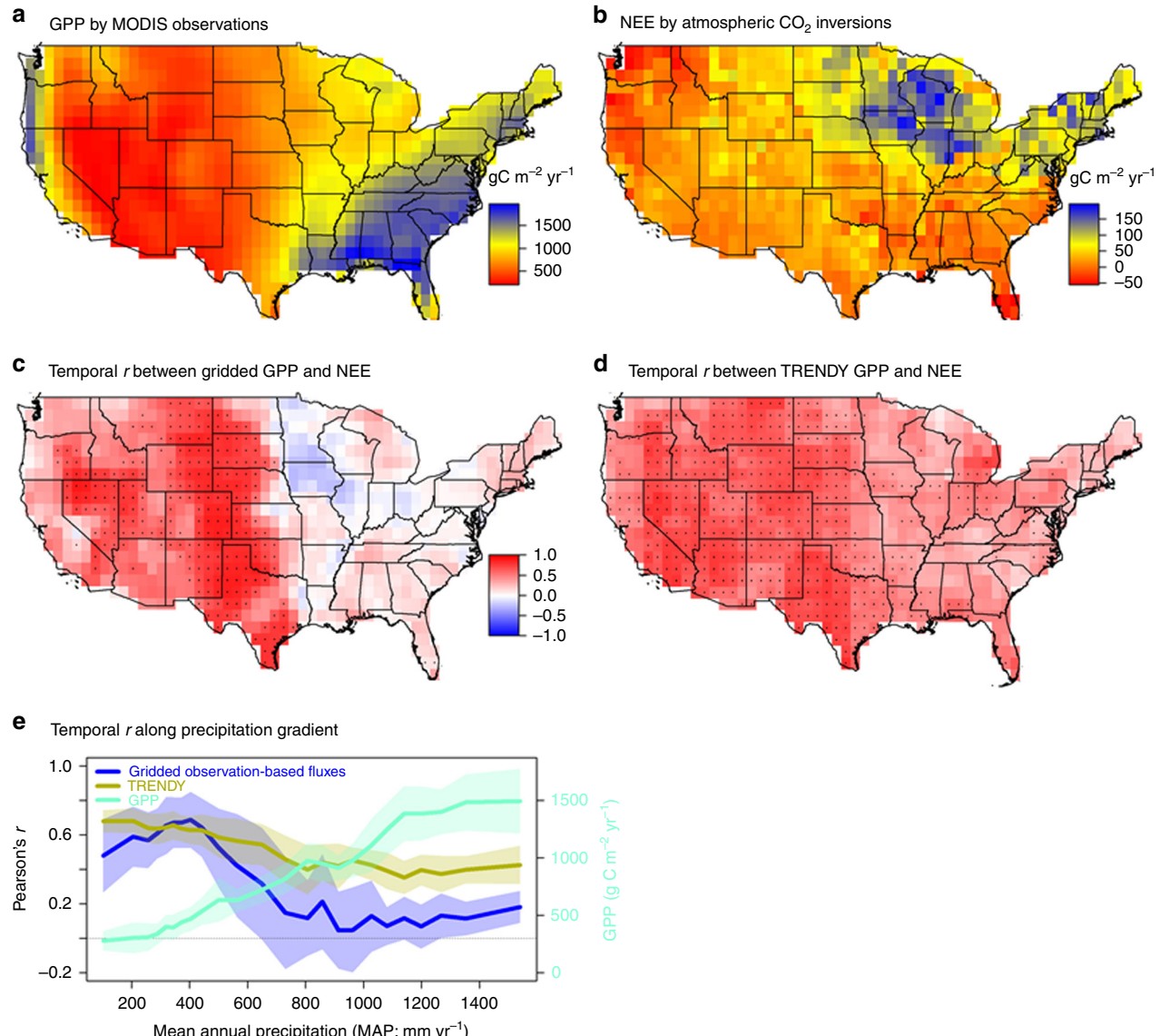

**Fig. 1** Relationship between production and net ecosystem carbon exchange along the precipitation gradient. Relationship between estimates of GPP (gross primary production) constrained by MODIS (Moderate Resolution Imaging Spectroradiometer) satellite observations and NEE (net ecosystem carbon exchange) constrained by atmospheric measurements in inversions varies along the precipitation gradient in the contiguous United States (CONUS). The spatial pattern of mean annual GPP (**a**, MOD17 GPP, 2000–2014) and NEE (**b**, ensemble mean from four atmospheric $CO_2$ inversion models between 2000 and 2014, a positive value indicates land as carbon sink) in the CONUS. **c** Mean temporal Pearson's $r$ between gridded observation-based GPP estimates ($n = 4$) and gridded NEE estimates ($n = 2$) between 2000 and 2014 (Supplementary Fig. 3). Dots indicate that correlation is significant if greater than four individual combinations are significant at 0.1 level in Supplementary Fig. 3. **d** Mean temporal Pearson's $r$ between modeled GPP and NEE by ten dynamic global vegetation models (DGVMs) from TRENDY project between 2000 and 2010 (Supplementary Fig. 15). Dots indicate a correlation is significant if more than five individual DGVM is significant at 0.1 level in Supplementary Fig. 15. **e** Mean Pearson's inter-annual $r$ between detrended GPP and NEE along the precipitation (and GPP) gradient in the CONUS, suggesting precipitation as the dominant control on GPP variations and GPP is the primary control on the inter-annual variation of NEE. Shaded areas are the mean ± one standard deviation within each precipitation bin. **a**–**d** were created in the R environment for statistical computing and graphics (https://www.r-project.org/)

this relationship is not an artifact of the observationally constrained dataset used. Wildfire $CO_2$ emission (Supplementary Fig. 5) and human activities, such as agriculture (Supplementary Fig. 6) also did not substantially change the temporal correlation between GPP and NEE along the precipitation gradient in the CONUS. In contrast, mean Pearson's $r$ between GPP and NEE from the ensemble of TRENDY DGVM simulations ($n = 10$ models) showed a universally positive temporal correlation across the CONUS that was stronger in more xeric western ecosystems (Fig. 1d). Mean Pearson's $r$ from TRENDY DGVM simulations became higher than that from observation-constrained estimates

in all regions where MAP is $> \sim 750\,\mathrm{mm\,yr^{-1}}$ (Fig. 1e). The varying strengths of GPP and TER controls on IAV of NEE along the continental precipitation gradient appear to be a robust pattern among observational datasets that is not well captured by DGVMs, in which NEE remains highly coupled to GPP across the entire CONUS.

**NEE sensitivity along the precipitation gradient.** To explore the underlying processes that control NEE sensitivity along the precipitation gradient, we compare the spatial and temporal

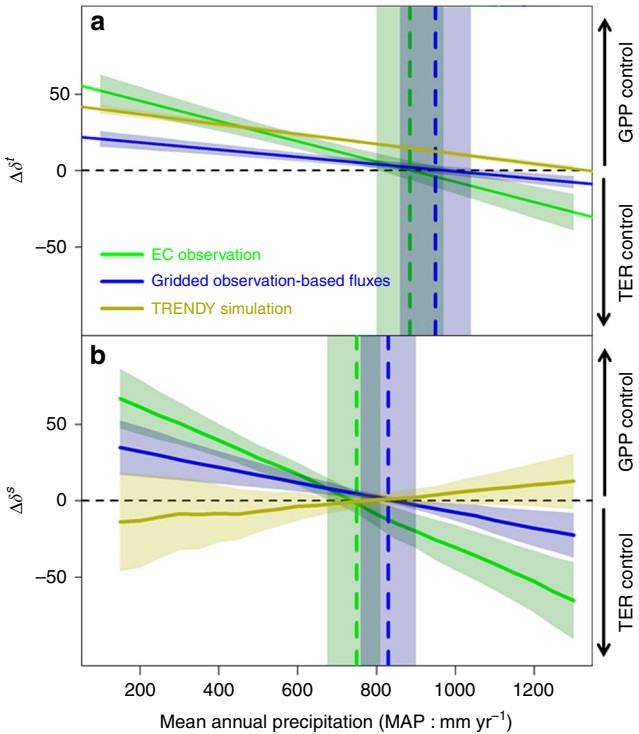

**Fig. 2** Sensitivity of ecosystem production and respiration to precipitation showed a threshold behavior. Sensitivity of ecosystem production (GPP) and respiration (TER) to precipitation showed a threshold behavior in both observation-constrained dataset (in green and blue), but not in TRENDY DGVM simulation (in olive). The Y axis ($\Delta\delta^t$ for temporal sensitivities or $\Delta\delta^s$ for spatial sensitivities in units of gC m$^{-2}$ yr$^{-1}$ in response to 100-mm change in precipitation) is calculated as the difference between the sensitivity of GPP to precipitation and the sensitivity of TER to precipitation for both temporal (**a**, $\Delta\delta^t = \delta^t_{GPP} - \delta^t_{TER}$) and spatial (**b**, $\Delta\delta^s = \delta^s_{GPP} - \delta^s_{TER}$) sensitivity. Because the sensitivity of TER ($\delta^t_{TER}$ or $\delta^s_{TER}$) to precipitation is always positive, a positive value ($\Delta\delta^t$ or $\Delta\delta^s$) indicates that GPP is more sensitive to precipitation than TER to precipitation anomalies. Shaded areas are the mean ± one standard deviation

sensitivity of production and respiration to climate controls. Regarding temporal sensitivities, we find that the IAV of both GPP and respiration is primarily driven by precipitation (Supplementary Fig. 7 and Supplementary Table 1) and thus focused our analysis on their sensitivities to precipitation (Fig. 2). Both observational datasets showed decreased IAV sensitivities of GPP ($\delta^t_{GPP}$ and $\delta^s_{GPP}$) and TER ($\delta^t_{TER}$ and $\delta^s_{TER}$) in response to increasing precipitation, but the slope of GPP sensitivity is steeper than TER (Supplementary Figs. 8–9 and Supplementary Table 1), which results in a precipitation threshold above which the IAV and local spatial gradients of NEE are controlled by GPP in more xeric ecosystems ($\Delta\delta^t > 0$ or $\Delta\delta^s > 0$) and by respiration in more mesic ecosystems ($\Delta\delta^t < 0$ or $\Delta\delta^s < 0$, Fig. 2). This precipitation threshold is the highest for temporal sensitivity using gridded observation-based fluxes (i.e., inversions of NEE and gridded GPP data products) ($\delta^t$: MAP = 950 ± 90 mm yr$^{-1}$, Fig. 2a) and lowest for spatial sensitivity using EC observations ($\delta^s$: 750 ± 75 mm yr$^{-1}$, Fig. 2b). The different precipitation thresholds between different observations may be due to data uncertainties in the large-scale gridded observation-based fluxes. The results also indicated that gridded observation-based fluxes, due to averaging out the ecosystem-scale variability, show lower sensitivity than in the EC observation (for both $\delta^t$ and $\delta^s$). Furthermore, the spatial sensitivity ($\delta^s$) is larger (a steeper slope) than the temporal sensitivity ($\delta^t$) (Fig. 2). This is likely because $\delta^s$ reflects gradients of different

vegetation types across precipitation, while $\delta^t$ only includes the short-term temporal response of fluxes to precipitation variability[14]. The legacy effect of previous year's precipitation on current-year's production may also contribute to the lower $\delta^t$[25,26]. The sensitivity of GPP to precipitation decreases from dry grass/ shrub ecosystem to wet forest ecosystems, also consistent with independent ecosystem-scale measurements[13,15].

Although TRENDY models also simulate the decreasing sensitivity of GPP and TER with increasing precipitation (Supplementary Fig. 10), they do not appear to show the same sensitivity threshold behavior than observation-based fluxes. The $\delta^t_{GPP}$ is always higher than $\delta^t_{TER}$ in the TRENDY models across the CONUS (Supplementary Fig. 10), and DGVMs appear to overestimate the sensitivity of GPP to precipitation in the more mesic ecosystems of CONUS. Thus, in more mesic ecosystems, DGVM simulations show that GPP is the dominant control on terrestrial NEE variability, while observationally constrained estimates show that TER is the dominant control on terrestrial NEE variability. This threshold in precipitation and model–data mismatch is also evident when looking at the fraction of precipitation being lost as evapotranspiration, indicating that water surplus may cause a shift in NEE variability to more respiration control (Supplementary Fig. 11).

**Potential mechanism for the data–model mismatch.** We then explore which ecosystem process—production or respiration— leads to this model–data mismatch. The per-pixel Pearson's temporal $r$ between GPP$_{MODIS}$ and TRENDY GPP (GPP$_{TRENDY}$) is universally positive in the CONUS (Supplementary Fig. 12), indicating that the mismatch is not likely due to GPP but rather due to TER in the more mesic ecosystems. Indeed, our results indicate that TER inverted from gridded observation-based fluxes (TER$_{inv}$ = GPP$_{MODIS}$−NEE$_{ACI}$) shows a significant temporal correlation with TRENDY TER (TER$_{TRENDY}$) in the more xeric ecosystems where MAP < 750 mm yr$^{-1}$ ($r = 0.72$, $p < 0.001$) (Fig. 3a), but less of a correlation in the more mesic ecosystems where MAP > 750 mm yr$^{-1}$ ($r = 0.302$, $p > 0.1$) (Fig. 3b). This suggests that DGVM TER simulations may be less realistic in more mesic regions than in more xeric regions and thus respiration most likely explains the mismatch between DGVMs and observations in more mesic ecosystems.

Total respiration is the sum of autotrophic respiration (Ra) by plants, and heterotrophic respiration (Rh) by soil microbes. The Rh, which composes about half of TER, is jointly controlled by carbon supply, soil properties, and by climate-dependent decomposition rates. We hypothesized that the relative influence of carbon supply versus environmental control on decomposition, especially soil moisture, over Rh is a function of water availability to drive the decoupling between GPP and TER, and thus varying the strength of temporal correlation between GPP and NEE along the precipitation gradient. To test this hypothesis, we used three simple empirical ecosystem respiration models with varying complexity and factors that included SOC (C), temperature (T), soil moisture (M), and current-year productivity (P) as a proxy of fresh input to litter in the fast SOC pools. These ecosystem respiration models aimed to identify key environmental factors that may help improve respiration simulation within DGVMs. We found that TER$_{TRENDY}$ and all empirical ecosystem respiration models were able to simulate the IAV of TER$_{inv}$ in the more xeric ecosystem ($r = 0.72$–0.869, $p < 0.01$), except for TER$_{CT}$ model ($r = -0.203$, $p > 0.05$) (Fig. 3a). By contrast, in the more mesic ecosystem, only the TER$_{CTMP}$ model captured the variance of TER$_{inv}$ ($r = 0.66$, $p < 0.001$) (Fig. 3b), and resulted in the lowest RMSE (Fig. 3c) and statistically indistinguishable estimate of TER when compared with TER$_{inv}$ (for the best TER$_{CTMP}$ model:

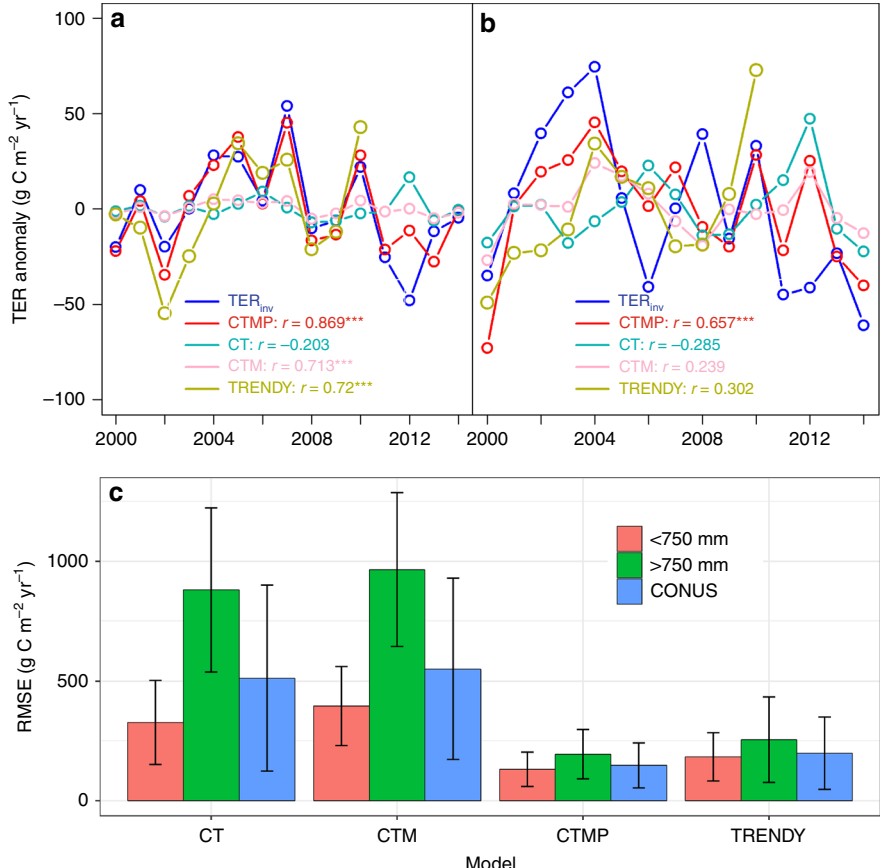

**Fig. 3** Ecosystem respiration was best explained by models that included production and soil carbon content, temperature and precipitation. Ecosystem respiration empirical model results showed that TER (total ecosystem respiration) inverted (TER$_{inv}$) from gridded observations of MODIS GPP (gross primary production) and inversion NEE (net ecosystem carbon exchange) was best explained by the TER$_{CTMP}$ model despite its larger number of degrees of freedom (AIC$_{CTMP}$ = 163.6, AIC$_{CT}$ = 242.8, and AIC$_{CTM}$ = 251.6, AIC stands for Akaike information criterion) that included GPP and soil carbon content, temperature and precipitation as predictors, especially in the humid region of CONUS. **a**, **b** are annual TER anomaly for regions below and above 750 mm yr$^{-1}$, respectively. The $r$ values in **a** and **b** showed the temporal correlation between TER$_{inv}$ anomaly and modeled TER anomaly. Symbols *,**,*** in **a** and **b** indicate the significant level at 0.1, 0.05, and 0.001 levels. **c** is the spatially averaged RMSE between TER$_{obs}$ and modeled TER (error bars show one standard deviation). CT: soil carbon/temperature model, CTM: soil carbon/temperature/water availability model, CTMP: soil carbon/temperature/water availability/productivity model

mean ± sd = 1211 ± 31 g C m$^{-2}$ yr$^{-1}$; TER$_{inv}$: 1242 ± 41 g C m$^{-2}$ yr$^{-1}$). Then, we partitioned TER$_{CMTP}$ into Ra being a GPP-dependent estimate, and Rh consisting of a GPP-dependent component standing for a fast-responding labile component of Rh and a GPP-independent term standing for Rh of slower soil carbon pools[27]. Comparison with an observed soil respiration database (SRDB v3)[28] confirms that the TER$_{CMTP}$ model performed better at simulating TER, Ra, and Rh than the DGVM simulations (Supplementary Fig. 13c). In particular, the TER$_{CMTP}$ simulations of Rh are much better at explaining the IAV of TER$_{inv}$ ($r = 0.44$, $p < 0.1$) than DGVM-simulated Rh ($r = 0.25$, $p > 0.1$) in more mesic regions. This suggests that DGVMs do not effectively simulate Rh and thus TER as water availability increases across the CONUS. Temporal correlation between IAV of gridded GPP$_{MODIS}$ and Rh from TER$_{CMTP}$ and TRENDY reveals that Rh may control the decoupling between GPP and NEE in the mesic CONUS areas (Fig. 4a, b).

Underestimation of the influence of soil moisture and soil carbon on Rh is a possible explanation of why DGVMs were not able to effectively simulate Rh in mesic ecosystems. DGVMs have routinely incorporated temperature and moisture constraints on Rh, but the effects of moisture on decomposition rate are much more uncertain than temperature, especially in warmer and wetter environments[29,30], and also soil-dependent[31]. Currently,

global land surface models like the ones in TRENDY appear to overestimate temperature effects on decomposition rates[29,32], and lead to faster soil carbon turnover time and stronger carbon–climate feedbacks[32,33], and therefore may override the influence of soil moisture and soil carbon on Rh[34]. In addition, experimental studies have shown that the sensitivity of microbial respiration to soil moisture increases in wetter ecosystems[35], and may contribute to the higher observed TER variability in the more mesic ecosystems. Partitioning TER$_{CMTP}$ into Rh and Ra confirms that Rh contributed more to IAV of TER than Ra in mesic regions than in the arid regions (0.50, calculated as S.D. of Rh divided by S.D. of TER$_{CMTP}$, vs. 0.40 in the drier region). Therefore, our analysis suggests that the model–data mismatch in more mesic ecosystems is most likely due to the poorly understood response of heterotrophic respiration to wetter conditions.

Large amounts of SOC are a major source of carbon supply for microbial decomposition[36], and can sustain Rh when the GPP anomalies are low and fresh labile carbon supply to Rh is being suppressed, therefore explaining why NEE variations tend to be more buffered against changes in GPP in more mesic ecosystems ($r = -0.22$, $p = 0.43$). Therefore, Rh may be more limited by environmental conditions, rather than by carbon supply in more mesic ecosystems, while Rh may be more limited by carbon

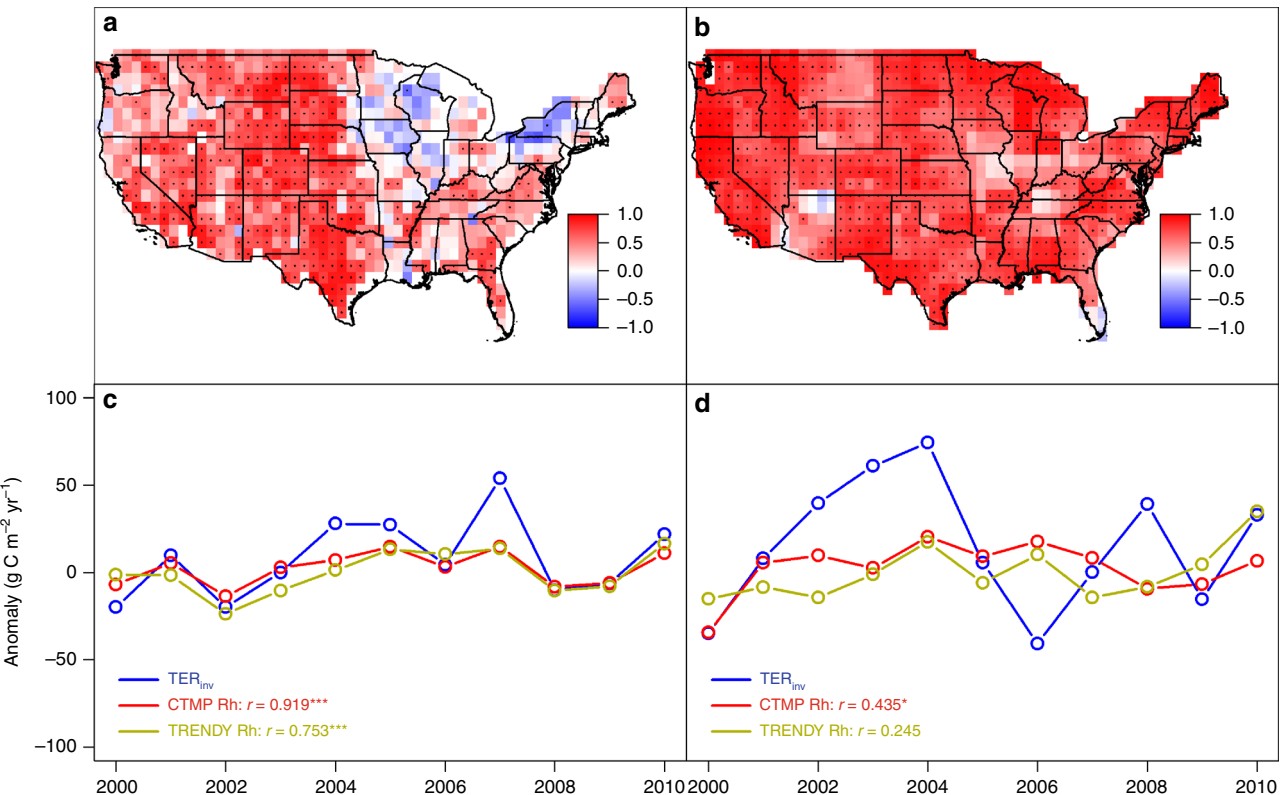

**Fig. 4** Temporal correlations between gridded production and respiration. Temporal correlations between gridded GPP and TER with estimates of Rh (heterotrophic respiration) derived from the TER$_{CTMP}$ empirical model. Satellite-derived estimates of GPP (gross primary production) are correlated with Rh predictions from the TER$_{CTMP}$ model (**a**) and TRENDY-derived estimates of GPP are correlated with Rh (**b**), and TER$_{inv}$ anomalies are correlated with TER$_{CTMP}$ Rh, and TRENDY Rh in regions below 750 mm yr$^{-1}$ (**c**) and above 750 mm yr$^{-1}$ (**d**). The correlation coefficients (r values) in **c** and **d** showed the temporal correlation between TER$_{inv}$ anomaly and model Rh anomaly. Symbols *,**,*** indicate significant level at 0.1, 0.05, and 0.001 levels. **a**, **b** were created in the R environment for statistical computing and graphics (https://www.r-project.org/)

supplied through productivity in drier ecosystems. Consistent with this interpretation, we see that SOC shows a similar increase across the continental precipitation gradient (Supplementary Fig. 14). Therefore, accurate simulation of Rh requires capturing the relative influence of carbon supply (e.g., production and SOC) and environmental constraints (e.g., soil moisture) under different hydrological conditions. However, GPP and Rh appear to be too tightly coupled in DGVM simulations in mesic ecosystems across the CONUS, suggesting that Rh may be overly dependent upon production in global land surface models, and therefore underestimate the influence of environmental constraints in the IAV of the Rh. This may help explain in part why land surface models tend to underestimate turnover times[27].

We also hypothesized that human land-use activity is a second plausible driver to cause the spatial mismatch between production and net carbon exchange in CONUS (Fig. 1a, b). The spatial inconsistence between annual mean GPP and NEE, especially in the Midwestern United States is most likely due to harvest of these intensive agricultural ecosystems. Agriculture statistics show that Midwest states account for about 21.2% of total agricultural land, but contributes to about 45.7% of crop export of the United States in 2012. The region exports roughly 0.08 Pg C yr$^{-1}$ of crop products[37], which is approximately half of the Midwest regional mean NEE (0.18 Pg C yr$^{-1}$) and one-third of CONUS mean NEE (0.3 Pg C yr$^{-1}$) between 2000 and 2014 in the region. The amount of carbon harvested in croplands needs to be better represented in DGVMs which lack realistic simulations of crop yields and often parameterize harvest as a fixed fraction of daily net primary production and do not consider the lateral transport

of C in harvested goods. Removal of crop yields also means reduced SOC inputs into the soil, and the production of harvest residues that will decompose faster than natural litter, e.g., because of tillage, should increase the control of NEE by Rh (Fig. 1c).

Our results help to understand the climate sensitivity of key carbon cycle processes and potential ways to improve DGVM simulations at the continental scale. Previous studies have shown that moisture-regulated productivity in the arid or semiarid region is the dominant control on IAV of global land net carbon exchange;[18,19] however, in these studies, they relied heavily on DGVM simulations that appear to be overly sensitive to the GPP response to water availability. To the extent that IAV is a useful diagnostic of long-term carbon–climate sensitivity, our results indicate that moisture-regulated respiration in mesic ecosystems can be another major mechanism regulating the variability of NEE. As the water balance of ecosystems within the United States is projected to be drier in certain regions and wetter in others[38], our analysis will facilitate the identification of potential critical thresholds which, if crossed, can abruptly change the carbon balance of ecosystems in CONUS. Our analysis highlights heterotrophic respiration as one of the most poorly understood carbon cycle processes and thus the most difficult to accurately simulate in land surface models, especially in more mesic ecosystems. Therefore, better understanding of the environmental controls of heterotrophic respiration may help improve carbon turnover times in model simulations, thereby reducing the amount of uncertainty in future carbon–climate feedbacks.

## Methods

**Temporal correlation between IAV of GPP and NEE**. We calculated the temporal correlation between IAV of detrended GPP and NEE using Pearson's product moment correlation (Pearson's $r$) at pixel level over the CONUS. For gridded observation-based fluxes, four gridded GPP or photosynthetic capacity indices combined with two gridded NEE estimates, resulting in eight individual Pearson's $r$ maps between IAV of GPP and NEE (Supplementary Fig. 3), were used to produce mean Pearson's $r$ (Fig. 1c) at 1° resolution from 2000 to 2014. For each pixel, if more than four individual correlations were significant, then the mean Pearson's $r$ is considered as significant. For TRENDY simulation, the mean Pearson's $r$ was calculated from detrended fluxes from each individual model ($n = 10$) at 1° resolution from 2000 to 2010 (Supplementary Fig. 14). For each pixel, if more than 5 individual correlations out of 10 TRENDY DGVM simulations were significant, then the mean Pearson's $r$ is considered as significant. Significance level at 0.1 was used in this study.

To test the robustness of temporal correlation between IAV of GPP and NEE, the same procedure was also applied at 3° spatial resolution and to growing-season fluxes (May–Oct) (Supplementary Fig. 3a, c, and d). To further reduce the uncertainties caused by pixel-level estimates for NEE$_{ACI}$, we also calculated the correlation between IAV of GPP observed by MODIS (GPP$_{MODIS}$) or TER$_{inv}$ and NEE$_{ACI}$ at the subcontinental scale (i.e., for the two regions with MAP above and below 750 mm yr$^{-1}$). The TER inverted from gridded observation-based fluxes (TER$_{inv}$) is calculated as the difference between GPP$_{MODIS}$ and NEE$_{ACI}$ (i.e., TER$_{inv}$ = GPP$_{MODIS}$–NEE$_{ACI}$). Although GPP$_{MODIS}$ and NEE$_{ACI}$ may be subject to large uncertainties at pixel level, we minimized this uncertainty and its influence on our main conclusion by using an ensemble mean from multiple data sources and regional estimates rather than pixel-level values (see Methods: Sensitivity analysis). For TRENDY simulations, GPP, TER, and NEE are calculated as ensemble mean of ten DGVMs from 2000 to 2010. We found a significant positive Pearson's $r$ between IAV of GPP and NEE in more xeric regions (MAP < 750 mm yr$^{-1}$) from both observations and TRENDY simulations, but correlations tended to diverge in the more mesic region (MAP > 750 mm yr$^{-1}$). TER$_{inv}$ and GPP$_{MODIS}$ have a correlation with NEE$_{ACI}$ of comparable absolute value but opposite sign in more mesic regions. In contrast, GPP still had a significant positive correlation with NEE in mesic regions in the TRENDY ensemble.

To test whether disturbance changes the temporal correlation between IAV of GPP and NEE, we compared the wildfire $CO_2$ emission with GPP$_{MODIS}$ and NEE$_{ACI}$, and found that wildfire $CO_2$ emissions are much smaller in magnitude ($0.021 \pm 0.0039$ Pg C yr$^{-1}$), compared to GPP$_{MODIS}$ ($6.29 \pm 0.26$ Pg C yr$^{-1}$) and NEE$_{ACI}$ ($0.30 \pm 0.13$ Pg C yr$^{-1}$) (Supplementary Fig. 5). To test whether human activities (e.g., agriculture) change the temporal correlation between IAV of GPP and NEE, we masked out regions with high human influence index (HII) with different thresholds (HII > 0.4 or HII > 0.3). We found that human activity had little influence on the temporal correlation between GPP and NEE along the precipitation gradient across the CONUS (Supplementary Fig. 6).

To visualize the temporal correlation between IAV of GPP and NEE along precipitation and GPP gradients, we plot the mean and standard deviation of Pearson's $r$ within each mean GPP$_{MODIS}$ and precipitation bins (Fig. 1e). Mean GPP and precipitation for each pixel was calculated from 2000 to 2014. We use precipitation from monthly, 0.5° spatial resolution from Climate Research Unit at the University of East Anglia. Mean annual GPP and precipitation was binned into 14 equal intervals. Mean and SD of mean Pearson's $r$ from constrained global observations (Fig. 1c), DGVM simulations (Fig. 1d), mean annual GPP, and mean annual precipitation (MAP) were summarized in each interval, and plotted along the GPP/precipitation gradient (Fig. 1e).

Finally, the soil organic carbon (SOC) content was plotted along the precipitation and GPP gradients to show its potential influence of the Pearson's $r$ in CONUS (Supplementary Fig. 14). The 0–100-cm SOC stock map was interpolated from measured SOC points by Rapid Carbon Assessment (RaCA) by the USDA-NRCS Soil Science Division in 2010 using Kriging method.

**Sensitivity analysis**. Temporal sensitivity ($\gamma^t_{flux}$ and $\delta^t_{flux}$): The temporal sensitivity was used to indicate the inter-annual sensitivity of carbon flux to change in climate factor for a given ecosystem over time. Therefore, the temporal sensitivity was calculated from each time-series measurement in which ecosystem production and respiration and climate factors have varied over time. Temporal relationship between ecosystem production and respiration and climate factors from long-term site-level data are usually modeled as linear regardless of ecosystem types[14]. In this analysis, the temporal model was formulated as

$$\Delta flux = \gamma^t_{flux}\Delta Temp + \delta^t_{flux}\Delta Prep \qquad (1)$$

where $\Delta$flux (i.e., GPP or TER), $\Delta$Temp, and $\Delta$Prep are annual anomalies for gross carbon flux, temperature, and precipitation, respectively. Therefore, $\gamma^t_{GPP}$ ($\gamma^t_{TER}$) and $\delta^t_{GPP}$ ($\delta^t_{TER}$) indicate the apparent temporal sensitivity of GPP (TER) to the absolute change ($\Delta$) of Temp and Prep controls. A summary for the temporal sensitivity was included in Supplementary Table 1, and the $\delta^t_{GPP}$ and $\delta^t_{TER}$ were used to generate Supplementary Fig. 2a and Supplementary Figs. 8–10. Annual anomalies ($\Delta$Temp and $\Delta$Prep) were calculated by removing the mean from the time-series data.

To calculate the relative contribution of Prep and Temp anomalies to the carbon flux anomalies (Supplementary Fig. 7), we follow the previous approach[20].

The product of a given sensitivity (e.g., $\delta^t_{GPP}$) and the corresponding climate-forcing anomaly (e.g., $\Delta$Prep) constitutes the flux anomaly component driven by this climate factor. Thus, $\Delta GPP = \gamma^t_{GPP}\Delta Temp + \delta^t_{GPP}\Delta Prep$ estimates the contributions of temperature ($\gamma^t_{GPP}\Delta Temp$) and precipitation ($\delta^t_{GPP}\Delta Prep$) anomalies to the carbon flux anomalies ($\Delta GPP$).

Spatial sensitivity ($\gamma^s_{flux}$ and $\delta^s_{flux}$): The spatial sensitivity was used to indicate the sensitivity of carbon fluxes across climate gradients (and ecosystem types). The spatial sensitivity was calculated from a spatially explicit gridded model and observation-based datasets. Spatial models are usually nonlinear between ecosystem production and respiration and climate factors when they span large gradients in climate[14]. We model the ecosystem production and respiration flux as a function of mean Temp and Prep using a polynomial function (up to two orders) to capture the nonlinear environmental effects.

$$flux = \alpha_0 + \alpha_1 Temp + \alpha_2 Temp^2 + \alpha_3 Prep + \alpha_4 Prep^2 \qquad (2)$$

Finally, the first-order derivative flux–climate curve is calculated as the spatial sensitivity of the flux to climate factors. We derived $\gamma^s_{GPP}$ ($\gamma^s_{TER}$) and $\delta^s_{GPP}$ ($\delta^s_{TER}$) to indicate the apparent spatial sensitivity and temporal sensitivities of GPP and TER to the change of Temp and Prep controls over space. A summary for the spatial sensitivity values is included in Supplementary Table 1, and the $\delta^s_{GPP}$ and $\delta^s_{TER}$ were used to generate Supplementary Fig. 2b and Supplementary Figs. 8–10.

Bootstrapping: to ensure that the sensitivity of ecosystem production and respiration to climate factors is not affected by extreme values, we performed 100 bootstrap analyses by randomly selecting a subset of data in each model. The confidence intervals of sensitivity in Fig. 2 and Supplementary Figs. 8–10 confirm that the threshold of ecosystem production and respiration to precipitation is not particularly sensitive to a few extreme values.

Sensitivity calculation for EC measurement: EC measurements provide direct observations of net ecosystem $CO_2$ exchange and estimated GPP and TER fluxes with climate variables. A total of 17 sites with at least 5 years of data, representing the major ecosystems across the CONUS were obtained from the FLUXNET2015 database (Supplementary Table 2 and Supplementary Fig. 16). Wetland sites and sites with recent major disturbance were excluded from our analyses. Daily GPP and TER were estimated as the mean value from both the nighttime partitioning method[39] and the light response curve method[40]. More details on the flux partitioning and gap-filling methods used are provided by ref. [41]. Daily values were summed to annual values, and then used to estimate the sensitivity of productivity (i.e., GPP) and respiration (i.e., TER) to annual Temp and Prep. The temporal sensitivity (i.e., $\gamma^t_{GPP}$, $\gamma^t_{TER}$, $\delta^t_{GPP}$, and $\delta^t_{TER}$, Supplementary Table 1) for each individual eddy-covariance site was calculated from time-series measurements and plotted along the precipitation gradient for each bootstrap replicate (Supplementary Fig. 8a). The spatial sensitivity (i.e., $\gamma^s_{GPP}$, $\gamma^s_{TER}$, $\delta^s_{GPP}$, and $\delta^s_{TER}$, Supplementary Table 1) was calculated from all 17 flux sites and plotted along the precipitation gradient (Supplementary Fig. 8b).

Sensitivity calculation for gridded observation-based fluxes: first, we used National Ecological Observatory Network (NEON) ecodomains to calculate spatial and temporal sensitivity of GPP and TER to Prep and Temp. There are 17 NEON ecodomains in the CONUS and these ecodomains were designed strategically to capture the variability in ecological and climatological conditions. Within each ecodomain, we summarize mean GPP$_{MODIS}$, NEE$_{ACI}$, TER$_{global\_obs}$, Prep, and Temp from 2000 to 2014. The temporal sensitivity (i.e., $\gamma^t_{GPP}$, $\gamma^t_{TER}$, $\delta^t_{GPP}$, and $\delta^t_{TER}$, Supplementary Table 1) for each individual NEON ecodomain was calculated from annual anomalies and plotted along the precipitation gradient for each bootstrap replicate (Supplementary Fig. 9a). The spatial sensitivity (i.e., $\gamma^s_{GPP}$, $\gamma^s_{TER}$, $\delta^s_{GPP}$, and $\delta^s_{TER}$ Supplementary Table 1) was calculated from long-term mean (2000–2014) gross carbon flux and climate of all 17 NEON ecodomains and plotted along the precipitation gradient (Supplementary Fig. 9b). The NEON ecodomains were obtained from (http://www.neonscience.org/data/maps-spatial-data).

The data uncertainties with GPP$_{MODIS}$ and NEE$_{ACI}$ may affect the spatial and temporal sensitivity of constrained global observations. The GPP$_{MODIS}$ uncertainty was mainly from its inputs (including MODIS observations of FPAR, LAI, land cover, and daily meteorological data) and algorithms[42]. Of these, meteorological data contribute to the largest uncertainty at the global scale, but this uncertainty is lower in regions with dense observations, such as CONUS[43]. Validation with EC measurement suggested that GPP$_{MODIS}$ shows reasonable spatial patterns and temporal variability across a diverse range of biomes and climate regimes[44]. The annual NEE$_{ACI}$ is the ensemble mean NEE of four atmospheric $CO_2$ inversions to reduce the uncertainty, primarily due to limited atmospheric data, uncertain prior flux estimates, and errors in the atmospheric transport models[45]. In North America, the largest uncertainty in NEE$_{ACI}$ is in the Midwestern United States, where agriculture dominates the landscape.

Sensitivity calculation for TRENDY simulation: The same procedure to calculate temporal and spatial sensitivity for constrained global observations is applied to the TRENDY simulations except (1) the temporal span is from 2000 to 2010; (2) GPP, NEE, and TER were the ensemble mean annual GPP and TER across ten DGVMs. The temporal and spatial sensitivity calculated from TRENDY simulations are plotted in Supplementary Fig. 10.

Comparing the climate sensitivity of GPP and TER along the precipitation gradient (Fig. 2). To compare the relative sensitivity of productivity and respiration to precipitation, we calculated the difference ($\Delta\delta^t$ or $\Delta\delta^s$) between the sensitivity of GPP to precipitation ($\delta^t_{GPP}$ and $\delta^s_{GPP}$) and sensitivity of TER to precipitation ($\delta^t_{TER}$ and $\delta^s_{TER}$) for each bootstrapping replicate (i.e., blue point minus red point in Supplementary Figs. 8–10 for each bootstrapping replicate). Because $\delta^t_{TER}$ and $\delta^s_{TER}$ were positive, a positive $\Delta\delta^t$ or $\Delta\delta^s$ indicates that GPP is more sensitive to precipitation than TER. Mean and 90 percentile of $\Delta\delta^t$ or $\Delta\delta^s$ ($n = 100$) was plotted along the precipitation gradient (Fig. 2). The $\Delta\delta^t$ or $\Delta\delta^s$ were summarized as a change in carbon flux (unit: gC m$^{-2}$ yr$^{-1}$) in response to 100-mm change in precipitation.

**Robustness of climate sensitivity of GPP and TER along the water availability gradient.** We used two other water availability indices, including mean annual precipitation minus evapotranspiration (P-ET, mm yr$^{-1}$) and the ratio between MAP and potential evapotranspiration (P/PET, unitless), to test the robustness of the climate sensitivity of GPP and TER along the water availability gradient. The P-ET integrates the temperature effect on water demand and is widely used to represent climate water deficit. The P/PET is an indicator of the degree of dryness of the climate at a given temperature. We calculated the sensitivity of GPP and TER to these two water availability indices for constrained global observation and TRENDY simulation (Supplementary Fig. 11). We did not report the sensitivity of GPP and TER to water deficit for constrained EC observation, as ET/PET was not included in the dataset. Monthly ET/PET data at 0.5° resolution were from MOD16 ET product (http://www.ntsg.umt.edu).

**Ecosystem respiration modeling experiment.** We designed a simple ecosystem respiration modeling experiment to diagnose why the DGVMs fail to capture the precipitation threshold of the sensitivity of production and respiration to precipitation. We used three empirical respiration models derived from publications with increasing complexity and factors that include observed SOC ($C$), temperature ($T$), soil moisture ($M$), and current-year production ($P$), and then compare them with TER$_{inv}$ (g C m$^{-2}$ yr$^{-1}$).

**TER$_{CT}$ model.** According to the models previously validated against a global database of soil respiration (Rs) observations[46], Rs can be predicted in response to soil $C$ content (SoilC, Mg ha$^{-1}$) and temperature (Temp, °C) as follows:

$$TER_{CT} = SoilC \times 64 \times 1.72^{0.21 \times Temp} \quad (3)$$

**TER$_{CTM}$ model.** On the basis of TER$_{CT}$ model, the effect of soil moisture (SoilM, m$^3$ m$^{-3}$) on Rs can be modeled as follows:[46]

$$TER_{CTM} = $$
$$SoilC \times 64 \times 1.72^{0.21 \times Temp} \times \left(\frac{SoilM - 2.1}{0.55 - 2.1}\right)^{6.6481} \times \left(\frac{SoilM + 0.007}{0.55 - 0.007}\right)^{3.23} \quad (4)$$

TER$_{CTMP}$ model: TER$_{CTMP}$ model is a photosynthesis-dependent respiration model that is calibrated and validated against eddy-covariance data[27]. TER$_{CTMP}$ combines the joint influences of temperature ($f(Temp)$), precipitation ($f(Prep)$, mm yr$^{-1}$), and substrate availability, including SOC (SoilC) and current-year production (P, g C m$^{-2}$ yr$^{-1}$), on ecosystem respiration, and can be described as follows:

$$TER_{CTMP} = (R_0 + k2 \times P) \times f(Temp) \times f(Prep) \quad (5)$$

where

$$R_0 = constant + a1 \times LAI_{max} + a2 \times SoilC \quad (6)$$

$$f(Temp) = e^{E_0 \times \left(\frac{1}{Tref - T0} - \frac{1}{Temp - T0}\right)} \quad (7)$$

$$f(P) = \frac{(a \times k + Prep \times (1 - a))}{(k + Prep \times (1 - a))} \quad (8)$$

In the TER$_{CTMP}$ model, $R_0$ is the reference respiration rate at the reference temperature (Tref) (15 °C), $E_0$ is the activation energy, and $T0 = -46.02$ °C. In the response of respiration to precipitation ($f(Prep)$), $k$ (mm) is the half-saturation constant of the hyperbolic relationship and $a$ is the response of total respiration to null Prep. LAImax is the maximum leaf area index within a pixel. LAI at 1-km$^2$ spatial resolution is derived from MODIS observations (MOD15A2, v6)[47]. Current-year GPP was used in the TER$_{CTMP}$ model as there is no evidence for lagged effects of GPP on TER$_{inv}$ or TER$_{CTMP}$ Rh (Supplementary Fig. 17). Conceptually, this model can be considered as the sum of a GPP-dependent term comprising autotrophic respiration (Ra) and the fast-responding labile component of heterotrophic respiration, and a GPP-independent term standing for

heterotrophic respiration (Rh) of slower carbon pools. Therefore, TER$_{CTMP}$ can be partitioned into Ra and Rh as follows:

$$Ra = k2 \times P \times f(Temp) \times f(Prep) \quad (9)$$

$$Rh = R_0 \times f(Temp) \times f(Prep) \quad (10)$$

All the coefficients used in TER$_{CTMP}$ were taken from the original study[27], where 104 globally distributed sites from the FLUXNET networks were used to derive plant functional-type specific parameters.

Model evaluation: using TER$_{inv}$ as a benchmark, we calculated the spatially averaged root-mean-squared error (RMSE) between four TER models (three empirical respiration models described above and one ensemble TRENDY TER (TER$_{TRENDY}$) and TER$_{inv}$ (Fig. 3c). We also calculated the temporal correlation between four TER models and TER$_{inv}$ at the subcontinental scale (MAP above and below 750 mm yr$^{-1}$) (Fig. 3a, b).

We also compared the TER$_{CTMP}$ and TER$_{TRENDY}$ with a global soil respiration database v3 (SRDB v3). Only measurements after 2000 were selected, and wetlands and deserts were excluded as well as disturbed samples (Supplementary Fig. 18). A total of 123 site-year data were used. Of all the SRDB v3 data, a total of 18 site-years explicitly measure the Rh and Ra, and these were selected to validate Ra and Rh from TER$_{CTMP}$ model and TRENDY simulations (Supplementary Fig. 13). Comparison between annual TER, Ra, and Rh from TER$_{CTMP}$ model and TRENDY DGVM simulations and the SRDB v3 showed that TER$_{CMTP}$ model explained significantly more variation in measured Rh in SRDB (v3) than DGVM simulations did (Supplementary Fig. 13c).

Temporal correlation between Rh derived from TER$_{CTMP}$ model and TRENDY simulations and GPP$_{MODIS}$ and GPP$_{TRENDY}$ were calculated at pixel level (Fig. 4a, b) and at the subcontinental scale (MAP above and below 750 mm yr$^{-1}$, Fig. 4c, d).

**Datasets.** Gridded observation-based fluxes. We used four remotely sensed observations of GPP or photosynthetic capacity indices, including MODIS 17 GPP (GPP$_{MODIS}$), solar-induced chlorophyll fluorescence (SIF), normalized difference vegetation index (NDVI), and fraction of photosynthetically active radiation (FPAR). The GPP$_{MODIS}$ is a product of maximum light-use efficiency, the FPAR, incoming radiation, and two scalar reduction factors that represent limitations on photosynthesis through temperature and vapor pressure deficit[42,48]. Monthly GPP$_{MODIS}$ at 0.05° resolution from 2000 to 2014 was obtained from the NTSG group (http://www.ntsg.umt.edu/). Annual mean GPP from 2000 to 2014 was used to produce Fig. 1a. SIF is sensitive to the electron transport rate of plant photosynthesis as well as the fraction of absorbed radiation[49,50], from the Global Ozone Monitoring Experiment-2 (GOME-2) for the period 2007–2014. The SIF data are retrieved near the $\lambda = 740$ nm far-red peak in chlorophyll fluorescence emission. Details of the retrieval of SIF from GOME-2 measurements can be found in ref. [49]. Monthly GOME-2 SIF at 0.05° resolution from 2007 to 2014 was used. NDVI is an index of landscape-integrated vegetation greenness and photosynthetic capacity, which is related to photosynthetic potentials under ideal environmental conditions, and thus NDVI reflects an inherent vegetation photosynthetic property. NDVI is from monthly, 0.05° MODIS MOD13C2 (C6) from 2000 to 2014, and only the data flagged as "good-quality" were used. FPAR is the fraction of absorbed photosynthetically active radiation that a plant canopy absorbs for photosynthesis and grows in the 0.4–0.7-μm spectral range. FPAR is from 8-day, 1-km resolution MODIS MCD15A2 (C5) from 2000 to 2014, and only the data flagged as "good-quality" were used. All the GPP or photosynthetic capacity indices were aggregated into an annual time step at 1° spatial resolution.

We used two gridded NEE estimates, including a NEE from an ensemble of four atmospheric CO$_2$ inversions (ACI) (NEE$_{ACI}$) and a NEE upscaled from eddy covariance flux data for North America (EC-MOD)[51]. Atmospheric CO$_2$ inversions estimate carbon exchange between the earth surface and atmosphere by utilizing atmospheric CO$_2$ measurements, a key observational component of the global carbon cycle (e.g., their observed temporal and spatial gradients). ACIs defer mainly because of choices for atmospheric observations, transport model, spatial and temporal flux resolution, prior fluxes, observation uncertainty and prior error assignment, and inverse method. Therefore, different ACI are likely different in spatial distribution and magnitude of carbon flux[45]. Four different ACI products, including Carbon-Tracker 2015 (CT2015)[52], Carbon-Tracker Europe 2015 (CTE2015)[53], CAMS[54], and Jena CarboScope v3.8[55,56], were obtained from 2000 to 2014, and resampled to 1° resolution using the nearest neighborhood at an annual time step. For each year from 2000 to 2014, an ensemble annual mean NEE was calculated across four ACIs (termed as NEE$_{ACI}$). Positive NEE indicates CO$_2$ from atmosphere to land ecosystem, and thus carbon sink for land ecosystem. Annual mean NEE$_{ACI}$ from 2000 to 2014 was used to plot Fig. 1b. The EC-NEE was developed from eddy covariance (EC) flux data, MODIS data streams, micrometeorological reanalysis data, stand age, and aboveground biomass data using a data-driven approach at the UNH[51]. EC-MOD NEE is obtained at 8-day time step at 1-km resolution between 2000 and 2012 and was aggregated into an annual time step at 1° spatial resolution. GPP and TER from the EC-MOD approach were not used, because they are not directly measured and inherently correlated with NEE.

TRENDY DGVM simulations. We used simulations of ten DGVMs from the TRENDY v2 ensemble[57] for the period 2000–2010: Hyland[58], JULES[59], LPJ[60], LPJ-GUESS[61], NCAR-CLM4[62], ORCHIDEE[63], OCN[64,65], SDVGM[66], and VEGAS[67]. The model ensemble stems from the TRENDY Inter-model Comparison ("Trends in net land_atmosphere carbon exchange over the period 1980_2010") that provided bottom-up estimates of carbon cycle processes for the Regional Carbon Cycle Assessment and Processes (RECCAP). Our analysis uses simulations from the "S2" storyline that includes time-varying atmospheric $CO_2$ concentrations and climate and fixed land cover for 2005. All simulations were based on climate forcing from the CRU-NCEPv4 climate variables at 6-h resolution for the years 1901–2010, including precipitation, snowfall, temperature, short-wave and long-wave radiation, specific humidity, air pressure, and wind speed. GPP, NEE, and TER were summarized at 1° spatial resolution at an annual timescale from 2000 to 2010 for each model.

EC observations. A total of 17 sites with at least 5 years of data, representing the major ecosystems across the CONUS were obtained from FLUXNET2015 database (Supplementary Table 2 and Supplementary Fig. 16). Consistent with $NEE_{ACI}$, positive NEE denotes uptake by the biosphere, and negative values indicate carbon losses.

Climate data. Monthly gridded temperature (Temp) and precipitation (Prep) at 0.5° spatial resolution from 2000 to 2014 were obtained from Climate Research Unit (CRU TS v. 3.25) at the University of East Anglia[68].

Global soil respiration database. The global soil respiration database (SRDB v3)[28] encompasses all published studies that report at least one of the following data measured in the field (not laboratory): annual total soil respiration (Rs), mean seasonal Rs, a seasonal or annual partitioning of Rs into its source fluxes (i.e., Ra and Rh), Rs temperature response (Q10), or Rs at 10° C from 1961 to 2012. In this analysis, we use records containing annual Rs (Ra or Rh, if present) after 2000 in the CONUS (Supplementary Fig. 18). Wetland and desert records were excluded. In total, we obtain 123 site-year annual Rs measurements, and 18 site-year Ra and Rh measurements.

Wildfire emission. Monthly gridded $CO_2$ emission from wildfire at 0.25° resolution is from global fire emission database (GFED4s, with small fires). Information about the algorithms, data, and uncertainties for the product can be found in ref. [69].

Human influence index. The human influence index (HII), an indictor of human impacts on the environment and ecosystem, was obtained from the Global Human Footprint Dataset of the Last of the Wild Project, Version 2, 2005 (LWP-2)[70]. The HII was created from nine global data layers covering human population pressure (population density), human land use and infrastructure (built-up areas, nighttime lights, and land use/land cover), and human access (coastlines, roads, railroads, and navigable rivers), and normalized by biome and realm.

## Data availability

All data analyzed in this study are publicly available. Gridded GPP by MODIS is obtained from the Numerical Terradynamic Simulation Group data portal (http://www.ntsg.umt.edu/data), NDVI and FPAR by MODIS is obtained the Land Processes Distributed Active Archive Center (LP DAAC: https://lpdaac.usgs.gov/), and SIF by GOME2 is obtained from NASA Aura Validation Data Center (AVDC) (https://avdc.gsfc.nasa.gov/pub/). TRENDY simulation is obtained from http://dgvm.ceh.ac.uk/index.html. Eddy-covariance data are from FLUXNET2015 Dataset (http://fluxnet.fluxdata.org/data/fluxnet2015-dataset/). Climate data are from Climate Research Unit (https://crudata.uea.ac.uk/cru/data/hrg/). Global soil respiration database (SRDB, v3) is obtained from ORNL DAAC (https://daac.ornl.gov/SOILS/guides/global_srdb_v3.html). Carbon tracker is obtained from NOAA Earth System Research Laboratory (https://www.esrl.noaa.gov/gmd/ccgg/carbontracker/), Carbon Tracker Europe from Wageningen University (http://www.carbontracker.eu/), Jena CarboScope is from MPG (http://www.bgc-jena.mpg.de/CarboScope/), and CAMS from ECMWF (http://apps.ecmwf.int/datasets/data/cams-ghg-inversions/).

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

## Acknowledgements

Z.L. was supported by NSF grant (1550932), NSFC (31470517), and CAS Pioneer Hundred Talents Program. W.R.L.A. acknowledges funding from the University of Utah Global Change and Sustainability Center, NSF Grant 1714972, and the USDA National Institute of Food and Agriculture, Agricultural and Food Research Initiative Competitive Programme, Ecosystem Services and Agro-ecosystem Management, grant no. 2017-05521. We thank Andrew R. Jacobson and John Miller for their helpful discussion on atmospheric $CO_2$ inversion models and data.

## Author contributions

Z.L. and A.P.B. conceived, designed the study, and led the manuscript writing. Z.L. analyzed the data. B.P., W.R.L.A., W.L., A.B. and P.C. contributed to the results inter- pretation and manuscript writing.

## Additional information

**Competing interests:** The authors declare no competing interests.

