## [Peer Review File · Nature Communications]

Reviewers' comments:

Reviewer #1 (Remarks to the Author):

General Comment

Plant productivity is generally regarded as the primary control of inter-annual variability of net-ecosystem-exchange (NEE). Through exhaustive analysis over the United States, where wealthy ecological data is available, authors found a threshold of annual precipitation, above which primary control of NEE switches from plant productivity to ecosystem respiration. Authors also showed that existing vegetation models fail to reconstruct such ecosystem behavior, probably due to dependency of heterotrophic respiration rate on soil wetness was not well reproduced in these models.

The design of analysis is fairly good and is well presented on the manuscript in general. Authors provided reasonable interpretations for the results. I have minor issues (mostly technical ones) on this manuscript as follows.

Specific concerns

Lines 48~52

Although this short abstract for Jung et al. plays a significant role for explaining motivations for this study, it is not clearly presented. Why their conclusions for local and global scales were different?

Line 81 "we built a suite of empirical ecosystem respiration models"

Line 172 "we constructed empirical ecosystem respiration models"

But in the line 383, authors state that "We used three empirical respiration models derived from publications".

Lines 87~88 "occupy different climate spaces"

Rephrasing or adding more explanation should be needed, because it's quite difficult to understand what this phrase means without seeing supplementary Figure S1.

Lines 88~89 "are not necessarily coherent spatially and temporally at regional scales"

I could not understand what it means.

Line 142~145

This logic for explaining different magnitudes of sensitivity between spatial and temporal climate variabilities is not clear and not enough.

Line 159

A type. Double periods.

Lines 182~184

Prior to this sentence, a short explanation is required for how TERCTMP was partitioned into Rh and Ra.

Line 213~214 "autotrophic respiration is dependent primarily on GPP"

This dependency is not obvious.

Autotrophic respiration can be partitioned into maintenance respiration and growth respiration. The former should be dependent primarily on biomass not on GPP. The latter can be independent on GPP when leaves are quickly emerging at the beginning of the growing season for example.

Lines 219~229

This paragraph stresses importance of agricultural activity on forming spatial coherence between

GPP and NEE. But, there are contradicting argument in lines 111–113.

Fig S12

No description for colors.

Reviewer #2 (Remarks to the Author):

This is an interesting paper on a topic of broad interest to the ecological and Earth System dynamics community. The question of how interannual variation in the foundational processes constructing landscape carbon exchange with the atmosphere are represented in models predicting future states of the Earth System and the temporal / spatial behavior of those processes informs contemporary grand challenges and our basic understanding of life processes. The authors identify a number of subsequent hypotheses that likely required extended work by the community (such as the relative supply versus environmental control over heterotrophic respiration as a function of mean annual precipitation). All-in-all, I think this effort is a worth contribution to the field and will likely influence a wide audience.

This work importantly extends site based, experimental, and more narrow treatment of the phenomena, incorporating model behavior and framing the question in the context of the continental categorization by the NEON effort. As such, it provides new context and new sets of ideas for the application of this knowledge and future work. In that regard, the work could be revised to be more engaging with a broader audience. While I identify a number of minor issues at the end of this review, three specific elements jump out to me where revision would help significantly in communicating ideas (and I will be clear here that this guidance is about the writing to a greater extent than the content. However, there is a chance to substantially improve impact by focusing on writing).

First, this approach of evaluating the sensitivity of a process (or set of process) by comparing and contrasting their spatial and temporal behavior has a history in the area of production that begins with Lauenroth and Sala 1992 Ecological Applications. It extends through the work of Paruelo and others, Knapp and Smith, Huxman and others, Bai and others (China), and current analyses focused on ecohydrology (Ponce-Campos and others). I think that it would help to both identify this history of questioning and to also explicitly identify to the reader early on that the current analysis is in the spirit of such space-time analyses that have been valuable at developing theory. It would allow the authors to more clearly identify their approach for analysis and bring the reader through the manuscript with a sense of direction.

Second, the descriptions / categorizations of the different regions of mean annual precipitation are not consistent, engaging, or necessarily true. In this analysis, all regions with MAP above an arbitrary threshold are termed, "mesic", whereas those falling below the threshold are "semi-arid". These terms are not antonyms and refer to different characteristics of landscapes. I would use simple terms such as 'more-mesic' and 'more-xeric', and then formally characterize the specifics when identifying explicit biomes for context. I would be careful to identify a number of context across the entire rainfall gradient rather than simply focusing on above or below the GPP-TER cross-over point. My concern here is that the findings will be extended more narrowly, or inappropriately to certain biomes without understanding that the terms were not more formally defined or that the simple message will be missed.

Finally, while I am certain that the use of multiple data streams (model and empirical) are critical for demonstrating the pattern of GPP versus TER sensitivity with respect to the interannual variability in NEE, but I find the manuscript difficult to read as a result of the introduced complexity from presenting a similar message over and over for these analyses. I think the manuscript would be much improved if the core message were clearly communicated (focusing on the ecosystem processes of interest; placing specific analyses in supplemental documents or

significantly reduced in scope) and reference were made to the consistency of the behavior in the multiple forms of analysis. Where there are discrepancies, they could then be more effectively highlighted to guide future hypothesis testing (such as the identification of the heterotrophic respiration hypothesis put forward). The alternative is a paper with a fairly straightforward message that is buried in repeated analysis with multiple proxy variables.

Some minor notes:

Throughout the manuscript, the word “productivity” is used when paired with “respiration” (as in “Ecosystem productivity and respiration”). Technically, “productivity” is a rate, while “production” is the process you are studying. Throughout, it would be much more appropriate to use “production” almost everywhere you currently use “productivity”.

“Decoupled” is an interesting word and I’m not sure it is consistently used with care. Part of the problem seems to be suggesting that you can directly compare GPP and NEE as processes that are sufficiently equivalent to evaluate in the context of a third process. Rather, of course as the authors point out and we all know, NEE is a function of GPP and TER. So to use the phrase term decouple as in the title is technically wrong (you are looking at how the variation in NEE relates to the variation in GPP or TER as a function of drivers).

Early in the text it would be helpful to have a short description of the kinds of data used. You use terms, like, “dense network of well constrained data”, and perhaps you could just say, “Ameriflux” or “CO2 flask data”?

The language on lines 143 and 144 could be firmed up – the use of the term ‘adaptation’ would be better replaced by more specific terms.

I think it would be much more appropriate to frame the analysis and discussion starting on lines 167 as a hypothesis emerging from the current analysis to explain the main findings (stating this more formally). The analysis presented is certainly quite compelling, but I can see a number of assumptions and tests that could be follow-ups that would be great contributions in future work.

The paragraph beginning on line 219 is slightly out of place and it would be helpful to try and be more specific about the purpose of this paragraph.

Reviewer #3 (Remarks to the Author):

In this paper, the authors address what might be an important issue in modelling terrestrial NEE at continental scales, the modelling of Rh response to changes in climate represented by those in temperature and precipitation. As in any large-scale study of this type, there is a large uncertainty in products such as GPP and NEE that are derived from large-scale products such as fpar and CO2 concentrations through models of GPP and atmospheric CO2 transport. Some discussion of this uncertainty, and how it might affect key findings of the paper, needs to be included when presenting the results, as noted on l. 63.

If ETP can be derived from fpar and temperature, then ETP:P ratios might be a more robust indicator of water limitation, and hence of the threshold between productivity and respiration controls on NEE, than would P alone as noted on l. 131. This ratio is often used in climate classification because water requirements rise sharply with temperature.

Might Rh be better correlated with productivity and hence litterfall from the previous year, as noted on l. 174?

I place little confidence in the predictive capabilities of models such as those used to explain variations in respiration, as noted on l. 387. However the need for a term for productivity to explain variation in TER of wetter climates was interesting, as noted on l. 174 and 197. The

authors should consider the implications of this finding for models that simulate Rh as first order functions of SOC. Some recent and current papers are showing that such models are less accurate than those using second order kinetics in which respiration is driven by microbial activity and so is more responsive to changes in C inputs. The authors should try to connect their findings about how IAV in TER is determined to how IAV in TER is modelled in current DGVMs. This connection would greatly increase the value of this paper.

Abstract

I. 27: specify how this control of NEE on IAV changes at this threshold, and why.

I. 30: How does this overestimation by DGVMs affect their modelling of the switch on IAV inferred from the observations?

Main Text

I. 38: its

I. 39: The authors should note that CO₂ assimilation drives ecosystem respiration at hourly to daily time scales through Ra and at seasonal to interannual time scales through litterfall.

I. 48: Both these hypotheses are likely valid under different climates, as noted later.

I. 63: These are all modelled products with large uncertainties. What are these uncertainties, and how do they affect estimates of GPP and NEE, and differences expressed as TER?

Results and Discussion

I. 92: Greatest NEP typically occur in temperate rainy climates with warm days that increase GPP and cool nights that reduce TER.

I. 131: Wouldn't P:ETp ratios be more informative than P to distinguish water-limited productivity, as P thresholds for water limitation would increase with temperature.

I. 160: TERobs is a misleading term, as both GPP and NEE are modeled products. TER is therefore a highly uncertain term derived from 2 modelled values, as noted in I.63 above.

I. 168: Ra has a different temperature sensitivity than does GPP and varies with biomass, so that the assumption of constant Ra:GPP is not likely to be valid.

I. 174: Residence times of different litterfall components may vary from < 1 to > 3 years, so that some lag in Rh after litterfall might be expected. What about testing Rh vs litterfall from the previous year?

I. 174: Results in Fig. 3 indicate that SOC is not a predictor of respiration rates in drier or wetter climates. Does this finding invalidate first-order SOC models? Does the need for productivity in the TER model for wetter climates indicate that litterfall rather than total SOC drives respiration? This is a key point for those models that still use first-order kinetics and could be developed further in this paper.

I. 183: Clarify the dependence of the TER models model on the TERobs data - i.e. this was a model fit, whereas the TRENDY results presumably were not.

I. 184 Fig. S12: (a) How can soil respiration exceed TER (b < 1)? (c) and (d): Are there enough points to derive significant conclusions (i.e. dF error = 3)?

I. 197: Do the TRENDY DGVMs calculate Rh from first order models of SOC which you have shown to be a poor predictor of TER? Do some use GPP which has a different dependence on temperature and precipitation? See comment on I. 174.

I. 212, Fig. S13: Check SOC values, they seem small for 0 - 100 cm

I. 216: ...overly dependent on current year's productivity. But what about the previous year's, as noted on I. 174. It might appear that DGVM R_h is too insensitive to productivity in mesic ecosystems, so that NEE varies too much with GPP.

Ecosystem respiration modeling experiments

I. 387: Include units of independent variables. I frankly place very little confidence in results from such models.

I. 389: This equation indicates a $Q_{10} > 3$, which seems large.

Reviewer #1 (Remarks to the Author):

General Comment

Plant productivity is generally regarded as the primary control of inter-annual variability of net-ecosystem-exchange (NEE). Through exhaustive analysis over the United States, where wealthy ecological data is available, authors found a threshold of annual precipitation, above which primary control of NEE switches from plant productivity to ecosystem respiration. Authors also showed that existing vegetation models fail to reconstruct such ecosystem behavior, probably due to dependency of heterotrophic respiration rate on soil wetness was not well reproduced in these models.

The design of analysis is fairly good and is well presented on the manuscript in general. Authors provided reasonable interpretations for the results. I have minor issues (mostly technical ones) on this manuscript as follows.

Specific concerns

[Comments 1.1] Lines 48~52

Although this short abstract for Jung et al. plays a significant role for explaining motivations for this study, it is not clearly presented. Why their conclusions for local and global scales were different?

[Response to comments 1.1] We clarified this from line 60-62. Specifically, we added the following explanation: "water-driven GPP and TER responses compensate each other, dampening water-driven NEE variability regionally, and therefore leaves a dominant temperature signal at global scale."

[Comments 1.2] Line 81 "we built a suite of empirical ecosystem respiration models"

Line 172 "we constructed empirical ecosystem respiration models"

But in the line 383, authors state that "We used three empirical respiration models derived from publications".

[Response to comments 1.2] We derived our empirical ecosystem respiration models from previous publications (see methods: 3. Ecosystem respiration modeling

experiment). To avoid confusion, we change these into “used” consistently through manuscript.

[Comments 1.3] Lines 87~88 "occupy different climate spaces"

Rephrasing or adding more explanation should be needed, because it's quite difficult to understand what this phrase means without seeing supplementary Figure S1.

[Response to comments 1.3] Here we mean the spatial pattern of mean annual GPP and NEE are not consistent with each other, and the highest GPP and NEE are in different climate regions. We have clarified this from line 100-108. “First, looking at spatial variations, ecosystem GPP is much more strongly controlled by mean annual precipitation (MAP; $r = 0.93$, $p < 0.001$) than by mean annual temperature (MAT; $r = 0.38$, $p < 0.001$), and increases with precipitation (Supplementary Fig. S2a), such that the highest mean annual GPP appears in the relatively warm and wet southeastern US (Fig. 1a). In contrast, the largest NEE (i.e., strong carbon sink) is found at intermediate levels of MAP ($\sim 750 - 1200 \text{ mm yr}^{-1}$), and then decreases at both higher MAP ($> 1200 \text{ mm yr}^{-1}$) and higher MAT ($> 20 \text{ }^\circ\text{C}$) (Supplementary Fig. S2b), such that the highest mean annual NEE appears in the relatively cool and wet North Central US (Fig. 1b).”

[Comments 1.4] Lines 88~89 "are not necessarily coherent spatially and temporally at regional scales"

I could not understand what it means.

[Response to comments 1.4] Here we mean that GPP and NEE are not consistent with each other in both spatial patterns and temporal variations. We have revised this into “find that mean annual GPP and NEE occupy different climate spaces (Supplementary Fig. S1) and do not necessarily co-vary spatially (Fig. 1a and b) or temporally (Fig 1c) at regional scales” (line 98 - 100).

[Comments 1.5] Line 142~145

This logic for explaining different magnitudes of sensitivity between spatial and temporal climate variabilities is not clear and not enough.

Response to reviewers

[Response to comments 1.5] We have explained why spatial sensitivity (δ^s) is larger than temporal sensitivity (δ^t) in greater detail from line 151-154. Specifically, “This is likely because δ^s reflects gradients of different vegetation types across precipitation, while δ^t only includes the short-term temporal response of fluxes to precipitation variability¹⁴. The “legacy effect” of previous year’s precipitation on current-year’s production may also contribute the lower δ^t ^{25,26}”

[Comments 1.6] Line 159

A type. Double periods.

[Response to comments 1.6] This has been corrected.

[Comments 1.7] Lines 182~184

Prior to this sentence, a short explanation is required for how TERCTMP was partitioned into Rh and Ra.

[Response to comments 1.7] We have adopted the suggestion and explained how TER_{CTMP} was partitioned into Ra and Rh from line 195-198. “Then, we partitioned TER_{CTMP} into Ra being a GPP-dependent estimate, and Rh consisting of a GPP-dependent component standing for fast-responding labile component of Rh and a GPP-independent term standing for Rh of slower soil carbon pools²⁷.”

[Comments 1.8] Line 213~214 "autotrophic respiration is dependent primarily on GPP"

This dependency is not obvious.

Autotrophic respiration can be partitioned into maintenance respiration and growth respiration. The former should be dependent primarily on biomass not on GPP. The latter can be independent on GPP when leaves are quickly emerging at the beginning of the growing season for example.

[Response to comments 1.8] Thank for pointing this out and reviewer #3 also raised the same comments [comments 3.14]. To avoid potential misleading statement, we have removed the related sentences, and focused on discussing Rh as the primary mechanism explaining the lack of correspondence between observed GPP and NEE along the precipitation gradient in the revised manuscript.

[Comments 1.9] Lines 219~229

This paragraph stresses importance of agricultural activity on forming spatial coherence between GPP and NEE. But, there are contradicting argument in lines 111~113.

[Response to comments 1.9] We apologize for this confusion.

This paragraph is meant to explain that human land use activity is a second plausible driver to cause the spatial mismatch between primary production and net carbon exchange (Fig 1a and b), while the lines 111~113 (original manuscript) cover that the temporal correlation between GPP and NEE along the precipitation gradient in the CONUS is robust.

To avoid the confusion (also address the Comments 2.10), we reframed this paragraph (Lines 219~229 in original manuscript) as a hypothesis, and added the following sentence from line 235-236. “We also hypothesized that human land-use activity is a second plausible driver to cause the spatial mismatch between production and net carbon exchange in CONUS (Fig 1a and b).”

The original sentences in lines 111~113 has been changed into “Wildfire CO₂ emission (Supplementary Fig S5) and human activities, such as agriculture, (Supplementary Fig S6) also did not substantially change the temporal correlation between GPP and NEE along the precipitation gradient in the CONUS” (line 123-125).

[Comments 1.10] Fig S12

No description for colors.

[Response to comments 1.10] The red line is TER_{CTMP} , and olive line is TRENDY simulation. We have clarified this (supplementary materials: line 153-154).

Reviewer #2 (Remarks to the Author):

This is an interesting paper on a topic of broad interest to the ecological and Earth System dynamics community. The question of how interannual variation in the foundational processes constructing landscape carbon exchange with the atmosphere are represented in models predicting future states of the Earth System and the temporal / spatial behavior of those processes informs contemporary grand challenges and our basic understanding of life processes. The authors identify a number of subsequent hypotheses that likely required extended work by the community (such as the relative supply versus environmental control over heterotrophic respiration as a function of mean annual precipitation). All-in-all, I think this effort is a worth contribution to the field and will likely influence a wide audience.

[Comments 2.1] This work importantly extends site based, experimental, and more narrow treatment of the phenomena, incorporating model behavior and framing the question in the context of the continental categorization by the NEON effort. As such, it provides new context and new sets of ideas for the application of this knowledge and future work. In that regard, the work could be revised to be more engaging with a broader audience. While I identify a number of minor issues at the end of this review, three specific elements jump out to me where revision would help significantly in communicating ideas (and I will be clear here that this guidance is about the writing to a greater extent than the content. However, there is a chance to substantially improve impact by focusing on writing).

[Response to comments 2.1] Thank for these excellent suggestions. These suggestions are very helpful in guiding our revision, and greatly improve the quality of our work. Please see our point-by-point response below.

[Comments 2.2] First, this approach of evaluating the sensitivity of a process (or set of process) by comparing and contrasting their spatial and temporal behavior has a history in the area of production that begins with Lauenroth and Sala 1992 Ecological Applications. It extends through the work of Paruelo and others, Knapp and Smith, Huxman and others, Bai and others (China), and current analyses focused on

ecohydrology (Ponce-Campos and others). I think that it would help to both identify this history of questioning and to also explicitly identify to the reader early on that the current analysis is in the spirit of such space-time analyses that have been valuable at developing theory. It would allow the authors to more clearly identify their approach for analysis and bring the reader through the manuscript with a sense of direction.

[Response to comments 2.2] This is an excellent suggestion. We now start our second paragraph by identifying the site-level studies on the climate sensitivity of production, followed by the explaining the competing hypotheses (e.g., water-controlled productivity vs. temperature-controlled respiration) on explaining global land carbon dynamics in different climate regions. Therefore, our study fills the gap between ecosystem-scale studies (for example, studies mentioned in this comment) and global-scale studies (for example, refs 3, 18, 19). Specifically, we added the following sentences from line 55-56. “A rich history of multi-year site-based data have revealed that the sensitivity of ecosystem production to precipitation decreases as water availability become more abundant¹²⁻¹⁷.”

The references mentioned here were also cited in the manuscript.

[Comments 2.3] Second, the descriptions / categorizations of the different regions of mean annual precipitation are not consistent, engaging, or necessarily true. In this analysis, all regions with MAP above an arbitrary threshold are termed, “mesic”, whereas those falling below the threshold are “semi-arid”. These terms are not antonyms and refer to different characteristics of landscapes. I would use simple terms such as ‘more-mesic’ and ‘more-xeric’, and then formally characterize the specifics when identifying explicit biomes for context. I would be careful to identify a number of context across the entire rainfall gradient rather than simply focusing on above or below the GPP-TER cross-over point. My concern here is that the findings will be extended more narrowly, or inappropriately to certain biomes without understanding that the terms were not more formally defined or that the simple message will be missed.

[Response to comments 2.3] We have adopted this suggestion, and used ‘more-mesic’ and ‘more-xeric’ throughout in the manuscript. To avoid the potential misinterpretation

due to this simple classification, we removed the specific biomes, and then also caution the reader that this is a regional-scale analysis, rather than a specific ecosystem.

[Comments 2.4] Finally, while I am certain that the use of multiple data streams (model and empirical) are critical for demonstrating the pattern of GPP versus TER sensitivity with respect to the interannual variability in NEE, but I find the manuscript difficult to read as a result of the introduced complexity from presenting a similar message over and over for these analyses. I think the manuscript would be much improved if the core message were clearly communicated (focusing on the ecosystem processes of interest; placing specific analyses in supplemental documents or significantly reduced in scope) and reference were made to the consistency of the behavior in the multiple forms of analysis. Where there are discrepancies, they could then be more effectively highlighted to guide future hypothesis testing (such as the identification of the heterotrophic respiration hypothesis put forward). The alternative is a paper with a fairly straightforward message that is buried in repeated analysis with multiple proxy variables.

[Response to comments 2.4] Thanks for the suggestion. To more clearly convey our main message and avoid confusion, we made the following efforts:

(1) We removed the repetitive description on the varying strength of temporal correlation between IAV of GPP and NEE along the precipitation gradient in the original manuscript.

(2) We removed the repetitive description of spatial and temporal sensitivity of GPP/TER along the precipitation gradient. Specifically, this description has been condensed into the following text (line 138-146). “Both observational datasets showed decreased IAV sensitivities of GPP (δ_{GPP}^t and δ_{GPP}^s) and TER (δ_{TER}^t and δ_{TER}^s) in response to increasing precipitation, but the slope of GPP sensitivity is steeper than TER (Supplementary Fig S8-S9 and Table S1), which results in a precipitation threshold above which the IAV and local spatial gradients of NEE are controlled by GPP in more xeric ecosystems ($\Delta\delta^t > 0$ or $\Delta\delta^s > 0$) and by respiration in more mesic ecosystems ($\Delta\delta^t < 0$ or $\Delta\delta^s < 0$, Fig. 2). This precipitation threshold is highest for temporal sensitivity using gridded observation-based fluxes (i.e., inversions NEE and gridded

GPP data-products) (δ^t : MAP = 950 ± 90 mm yr⁻¹, Fig 2a) and lowest for spatial sensitivity using EC observations (δ^s : 750 ± 75 mm yr⁻¹, Fig 2b).”

(3) We also shortened the description of the ecosystem respiration model results.

We believed the main message of the manuscript is more clearly communicated and reader will have a better understanding of the analysis and takeaways in the revised manuscript.

Some minor notes:

[Comments 2.5] Throughout the manuscript, the word “productivity” is used when paired with “respiration” (as in “Ecosystem productivity and respiration”). Technically, “productivity” is a rate, while “production” is the process you are studying. Throughout, it would be much more appropriate to use “production” almost everywhere you currently use “productivity”.

[Response to comments 2.5] Yes, we have changed productivity into production in the manuscript.

[Comments 2.6] “Decoupled” is an interesting word and I’m not sure it is consistently used with care. Part of the problem seems to be suggesting that you can directly compare GPP and NEE as processes that are sufficiently equivalent to evaluate in the context of a third process. Rather, of course as the authors point out and we all know, NEE is a function of GPP and TER. So to use the phrase term decouple as in the title is technically wrong (you are looking at how the variation in NEE relates to the variation in GPP or TER as a function of drivers).

[Response to comments 2.6] To avoid the confusion and highlight the main conclusion of the analysis, we have changed to title into “Precipitation thresholds regulate net carbon exchange at the continental scale”.

[Comments 2.7] Early in the text it would be helpful to have a short description of the kinds of data used. You use terms, like, “dense network of well constrained data”, and perhaps you could just say, “An Comparison between annual TER eriflux” or “CO2 flask data”?

[Response to comments 2.7] Thank for the suggestion, we have added short descriptions of the kinds of data used in abstract from line 24-27. Specifically “Here we investigate the climate sensitivity of processes controlling the interannual variability (IAV) of net ecosystem carbon exchange (NEE) across the contiguous United States (CONUS) using a dense array of atmospheric, satellite and eddy-covariance observations.”

[Comments 2.8] The language on lines 143 and 144 could be firmed up – the use of the term ‘adaptation’ would be better replaced by more specific terms.

[Response to comments 2.8] This has been addressed in response to review #1. Please see [Response to comments 1.5]

[Comments 2.9] I think it would be much more appropriate to frame the analysis and discussion starting on lines 167 as a hypothesis emerging from the current analysis to explain the main findings (stating this more formally). The analysis presented is certainly quite compelling, but I can see a number of assumptions and tests that could be follow-ups that would be great contributions in future work.

[Response to comments 2.9] This is an excellent suggestion.

To explain the mechanism underlying the varying strength of temporal correlation between GPP and NEE, We hypothesized that the relative influence of carbon supply versus environmental control on decomposition, especially soil moisture, over Rh is a function of water availability to drive the decoupling between GPP and TER, and thus varying strength of temporal correlation between GPP and NEE along precipitation gradient (line 181-185).

Response to reviewers

To explain the spatial mismatch between mean annual GPP and NEE, We also hypothesized that human land-use activity is a second plausible driver to cause the spatial mismatch between production and net carbon exchange in CONUS (Fig 1a and b).(line 235-236)

[Comments 2.10] The paragraph beginning on line 219 is slightly out of place and it would be helpful to try and be more specific about the purpose of this paragraph.

[Response to comments 2.10] Thanks for suggestion. This paragraph is meant to explain the spatial mismatch between mean annual GPP and NEE. We have clearly stated the objective of the paragraph, and presented as hypotheses (line 235-236).

“We also hypothesized that human land-use activity is a second plausible driver to cause the spatial mismatch between production and net carbon exchange in CONUS (Fig 1a and b).”

.

Reviewer #3 (Remarks to the Author):

[Comments 3.1] In this paper, the authors address what might be an important issue in modelling terrestrial NEE at continental scales, the modelling of Rh response to changes in climate represented by those in temperature and precipitation. As in any large-scale study of this type, there is a large uncertainty in products such as GPP and NEE that are derived from large-scale products such as fpar and CO2 concentrations through models of GPP and atmospheric CO2 transport. Some discussion of this uncertainty, and how it might affect key findings of the paper, needs to be included when presenting the results, as noted on l. 63.

[Response to comments 3.1] Thanks for the comments.

We have now improved our discussion of the data uncertainties with GPP and NEE_{ACI} . Specifically, we added the following discussion from line 390-400. “The data uncertainties with GPP_{MODIS} and NEE_{ACI} may affect the spatial and temporal sensitivity of constrained global observations. The GPP_{MODIS} uncertainty was mainly from its inputs (including MODIS observations of FPAR, LAI, land cover, and daily meteorological data) and algorithm⁴³. Of these, meteorological data contributes the largest uncertainty at global scale, but this uncertainty is lower in regions with dense observations, such as CONUS⁴⁴. Validation with EC measurement suggested that GPP_{MODIS} shows reasonable spatial patterns and temporal variability across a diverse range of biomes and climate regimes⁴⁵. The annual NEE_{ACI} is the ensemble mean NEE of four atmospheric CO2 inversions to reduce the uncertainty, primarily due to limited atmospheric data, uncertain prior flux estimates and errors in the atmospheric transport models⁴⁶. In North America, the largest uncertainty in NEE_{ACI} is in the in the Midwestern US, where agriculture dominates the landscape.”

We believe data uncertainties with GPP and NEE_{ACI} may contribute to the different precipitation thresholds compared with EC measurement. We clarified this from line 146-148. “The different precipitation thresholds between different observations may be due to data uncertainties in the large-scale gridded observation-based fluxes.”

Realizing data uncertainty may affect our analysis and conclusion; we used the following ways to minimize the potential influence of data uncertainty on our major conclusions. (1) To ensure the temporal correlation between GPP and NEE is robust, we reported the ensemble mean of 8 correlation resulting from four independent GPP proxies and two independent NEE estimate. We also assessed the potential influence of spatial resolution or season (Supplementary Fig. S4), and other factors including wildfire CO₂ emission (Supplementary Fig S5) and human activities, such as agriculture, (Supplementary Fig S6). (2) To ensure the precipitation threshold behavior of relative sensitivity of production and respiration to water availability is robust, we used two different observation-constrained datasets, including constrained EC measurement and constrained global observation (e.g., GPP by MODIS observation and NEE by atmospheric CO₂ inversion). (3) To reduce pixel-level uncertainties NEE by atmospheric CO₂ inversion, we used the ensemble mean NEE from four atmospheric CO₂ inversion and summarized the NEE at the 17 NEON domains across the CONUS. (4) In addition, we also tested the threshold behavior using different water availability indices, including ratio between precipitation and potential evapotranspiration and precipitation minus actual evapotranspiration (Supplementary Fig S11). Our additional analysis suggested the threshold behavior of carbon cycle along the precipitation is a robust behavior, regardless of water availability indices used.

In short, while important to acknowledge and highlight for readers, we have concluded that data uncertainties will likely not change the main findings of this analysis.

[Comments 3.2] If ETP can be derived from f_{par} and temperature, then $ET_p:P$ ratios might be a more robust indicator of water limitation, and hence of the threshold between productivity and respiration controls on NEE, than would P alone as noted on l. 131. This ratio is often used in climate classification because water requirements rise sharply with temperature.

[Response to comments 3.2] This is a great idea. We conducted additional analysis using (1) $P-ET$, (2) $P:ET_p$ as alternative water availability indicator. We also find a threshold behavior from constrained global observations, but not the TRENDY

simulation (Supplementary Fig S11). This demonstrated that our results are robust, regardless of the water availability indices used. Actually, these water availability indices are highly correlated with precipitation both spatially and temporally. In fact, most of previous analysis (e.g., studies mentioned by review 2) used precipitation as water availability indicator. We use precipitation also because it is the most widely measured, readily available over large spatiotemporal scales, and considered as the most direct indicator of future climate factor.

Specifically, we added the following text from line 165-167. “This threshold in precipitation and model-data mismatch is also evident when looking at the fraction of precipitation being lost as evapotranspiration, indicating that water surplus may cause a shift in NEE variability to more respiration control (Supplementary Fig. S11).” We also clarified why we chose precipitation as water availability indicator from line 82 – 85. “Precipitation is a simple measure of ecosystem water availability that is accurately measured across CONUS, and there are several lines of evidence for strong relationships with ecosystem production in this region¹⁵.”

We added a section named “2.4 Robustness of climate sensitivity of GPP and TER along the water availability gradient” to highlight this additional analysis. Specifically, we added the following text from line 416-426.

“2.4 Robustness of climate sensitivity of GPP and TER along the water availability gradient

We used two other water availability indices, including mean annual Precipitation minus Evapotranspiration (P-ET, mm yr⁻¹) and the ratio between MAP and potential Evapotranspiration (P/PET, unitless), to test the robustness of the climate sensitivity of GPP and TER along the water availability gradient. The P-ET integrates the temperature effect on water demand and is widely used to represent climate water deficit. The P/PET is an indicator of the degree of dryness of the climate at a given. We calculated the sensitivity of GPP and TER to these two water availability indices for constrained global observation and TRENDY simulation (Supplementary Fig S11). We did not report the sensitivity of GPP and TER to water deficit for constrained EC observation, as ET/PET was not included in the dataset. Monthly ET/PET data at 0.5°

resolution was from MOD16 ET product (<http://www.ntsg.umd.edu>). ”

[Comments 3.3] Might Rh be better correlated with productivity and hence litterfall from the previous year, as noted on l. 174?

[Response to comments 3.3] We acknowledged that Rh may be correlated with litterfall at various time lag (~ months - ~ years) at ecosystem levels. However, it is difficult for us to obtain litterfall production and apply a time lag across the CONUS in the ecosystem respiration models. Instead, we compared the GPP and TER/Rh in the region above and below 750 mm yr^{-1} , and it does not appear there is a lagged effect of GPP on total respiration or Rh (from TER_{CTMP}) at regional scale (Supplementary Fig S17). We have clarified this from line 456-457 (see method: 3. Ecosystem respiration modeling experiments).

[Comments 3.4] I place little confidence in the predictive capabilities of models such as those used to explain variations in respiration, as noted on l. 387. However the need for a term for productivity to explain variation in TER of wetter climates was interesting, as noted on l. 174 and 197. The authors should consider the implications of this finding for models that simulate Rh as first order functions of SOC. Some recent and current papers are showing that such models are less accurate than those using second order kinetics in which respiration is driven by microbial activity and so is more responsive to changes in C inputs. The authors should try to connect their findings about how IAV in TER is determined to how IAV in TER is modelled in current DGVMs. This connection would greatly increase the value of this paper.

[Response to comments 3.4]

In this analysis, we used empirical ecosystem respiration models with varying complexity. The main objective for these ecosystem respiration models is to identify key environmental factors that may help improve respiration simulation within DGVMs, rather than accurate simulation of respiration (line 188-189).

A key finding for the ecosystem respiration model experiment suggested that the relative influence of carbon supply versus environmental control, especially soil moisture, over R_h is a function of water availability along precipitation gradient (as pointed by reviewer 2). As shown in Fig 3, the respiration model without production (e.g., TER_{CT}/TER_{CTM}) has lower interannual variability (IAV) of respiration in the more xeric ecosystem as respiration is more controlled by SOC which is relative stable among years (fig 3a). While there is high IAV of respiration in the more mesic ecosystem as respiration is more controlled by environmental constraints which varies a lot among years (fig 3b). The TER_{CTMP} seems to capture the relative influence of carbon supply versus environmental control along the precipitation gradient. Therefore, accurate simulation of R_h requires capturing the relative influence of carbon supply (e.g., production and SOC) and environmental constraint (e.g., soil moisture) under different hydrological conditions. However, GPP and R_h appear to be too tightly coupled in DGVM simulations in mesic ecosystems across the CONUS, suggesting that R_h may be overly dependent upon production in global land surface models, and therefore underestimate the influence of environmental constrain in the IAV of the R_h . (Line 227-233).

Abstract

[Comments 3.5] I. 27: specify how this control of NEE on IAV changes at this threshold, and why.

[Response to comments 3.5] “Here, we investigate the climate sensitivity of processes controlling the interannual variability (IAV) of net ecosystem carbon exchange (NEE) across the contiguous United States (CONUS) using a dense array of atmospheric, satellite and eddy-covariance observations. We show a precipitation threshold between 750 – 950 mm yr⁻¹, below which the IAV of NEE is regulated by ecosystem production and above which IAV of NEE appears to be regulated by ecosystem respiration. This precipitation threshold is evident across multiple datasets and observation scales indicating that it is a robust result and a possible emergent constraint for evaluating models.” We have clarified this from line 25-32.

[Comments 3.6] l. 30: How does this overestimation by DGVMs affect their modelling of the switch on IAV inferred from the observations?

[Response to comments 3.6] The DGVM models overestimate the sensitivity of production to water availability, and therefore simulated GPP is the primary control on IAV of NEE across the whole precipitation gradient within the CONUS.

Main Text

[Comments 3.7] l. 38: its

[Response to comments 3.7] this has been corrected.

[Comments 3.8] l. 39: The authors should note that CO₂ assimilation drives ecosystem respiration at hourly to daily time scales through Ra and at seasonal to interannual time scales through litterfall.

[Response to comments 3.8] We have clarified this point. “Terrestrial NEE represents the small imbalance between CO₂ assimilation through gross primary production (GPP) and CO₂ release through total ecosystem respiration (TER). GPP and TER are coupled over the long-term through the distribution of carbon assimilated to ecosystem carbon pools and their subsequent turnover leading to TER. Yet, GPP and TER can be decoupled on temporal scales going from years to centuries if one of these fluxes is perturbed by environmental conditions and small decoupled variations in GPP or TER fluxes can result in large variations in NEE.”(line 45-51).

[Comments 3.9] l. 48: Both these hypotheses are likely valid under different climates, as noted later.

[Response to comments 3.9] Thanks for the comment.

[Comments 3.10] l. 63: These are all modelled products with large uncertainties. What

are these uncertainties, and how do they affect estimates of GPP and NEE, and differences expressed as TER?

[Response to comments 3.10] please see our [Response to comments 3.1]

Results and Discussion

[Comments 3.11] l. 92: Greatest NEP typically occur in temperate rainy climates with warm days that increase GPP and cool nights that reduce TER.

[Response to comments 3.11]. Thanks for the comments. Here we discussed the spatial pattern of mean NEE in the CONUS. We found the highest mean annual NEE appears in the relative cool and wet North Central US. We have clarified this from line 104-108. “In contrast, the largest NEE (i.e., strong carbon sink) is found at intermediate levels of MAP ($\sim 750 - 1200 \text{ mm yr}^{-1}$), and then decreases at both higher MAP ($> 1200 \text{ mm yr}^{-1}$) and higher MAT ($> 20 \text{ }^\circ\text{C}$) (Supplementary Fig. S2b), such that the highest mean annual NEE appears in the relatively cool and wet North Central US (Fig. 1b)”

[Comments 3.12] l. 131: Wouldn't P:ETp ratios be more informative than P to distinguish water-limited productivity, as P thresholds for water limitation would increase with temperature.

[Response to comments 3.12] please see [Response to comments 3.2].

[Comments 3.13] l. 160: TERobs is a misleading term, as both GPP and NEE are modeled products. TER is therefore a highly uncertain term derived from 2 modelled values, as noted in l.63 above.

[Response to comments 3.13] Thank for the comments. To avoid the confusion, we changed the MODIS GPP ($\text{GPP}_{\text{MODIS}}$) and NEE by atmospheric CO₂ inversion (NEE_{ACI}) into gridded observation-based fluxes because they are partially constrained by the observation (e.g., MODIS satellite observations and atmospheric CO₂ measurement). Therefore, TERobs has been changed to TER_{inv} as it is inverted from gridded

observation-based fluxes. We updated the terminology throughout the revised manuscript. All the figures were also updated accordingly in the revised manuscript.

[Comments 3.14] l. 168: Ra has a different temperature sensitivity than does GPP and varies with biomass, so that the assumption of constant Ra:GPP is not likely to be valid.

[Response to comments 3.14] This issue is also raised by reviewer 1. Please see [Response to comments 1.8].

[Comments 3.15] l. 174: Residence times of different litterfall components may vary from < 1 to > 3 years, so that some lag in Rh after litterfall might be expected. What about testing Rh vs litterfall from the previous year?

[Response to comments 3.15] Please see our [Response to comments 3.3].

[Comments 3.16] l. 174: Results in Fig. 3 indicate that SOC is not a predictor of respiration rates in drier or wetter climates. Does this finding invalidate first-order SOC models? Does the need for productivity in the TER model for wetter climates indicate that litterfall rather than total SOC drives respiration? This is a key point for those models that still use first-order kinetics and could be developed further in this paper.

[Response to comments 3.16] In this study, our statistical empirically-calibrated ecosystem respiration model (i.e., TER_{CTMP}) suggested that current-year production is an important carbon supply for soil respiration, and improves the simulation of interannual variation respiration. The widely used mechanistic process model of Rh (e.g., CENTURY type model or its variants in the DGVMs) used first-order SOC models at much finer temporal resolution (e.g., daily - monthly). Therefore, the scale difference between our statistical (annual time step) and first-order SOC models (daily to monthly time step) makes it difficult for direct comparison.

However, our analysis did suggest that including production as additional carbon supply, which is fast-responding labile component of heterotrophic respiration, corresponding to the LAI_{max} part of TER_{CTMP} model (line 448), can increase the variability explained by Rh, since SOC is relative stable and abundant in the wetter region. Therefore, including

productivity into first-order SOC models in the TER model for wetter climate region may increase the Rh estimate, but remain to be tested.

Please also see our [Response to comments 3.4].

[Comments 3.17] l. 183: Clarify the dependence of the TER models model on the TERobs data - i.e. this was a model fit, whereas the TRENDY results presumably were not.

[Response to comments 3.17] Here we showed that TER_{CTMP} model is better at simulating TER, Ra, Rh from SRDB, as an independent evaluation of the TER_{CTMP} model. The TER models (e.g., TER_{CT} , TER_{CTM} , TER_{CTMP}) are from independent datasets, and therefore not related to TER_{inv} .

[Comments 3.18] l. 184 Fig. S12: (a) How can soil respiration exceed TER ($b < 1$)? (c) and (d): Are there enough points to derive significant conclusions (i.e. dF error = 3)?

[Response to comments 3.18]

(a) The most possible reason that SRDB Rs is a field measurement with a relatively small footprint while the modeled TER (y axis, TER_{CTMP}) represented by an average estimate at 1 degree. We have clarified this in the line 150 (supplementary materials).

(c and d): We agree that there is only a small number of samples ($n = 5$) and may violate the homogeneity assumption for the parametric test. Therefore, we used a non-parametric Spearman's rho test, which does not require the norm distribution of the dataset, in the revised manuscript. The Spearman's rho test also suggested that Rh from TER_{CTMP} model is better at simulating Rh than TRENDY models (supplementary materials: line 151-153).

[Comments 3.19] l. 197: Do the TRENDY DGVMs calculate Rh from first order models of SOC which you have shown to be a poor predictor of TER? Do some use GPP which has a different dependence on temperature and precipitation? See comment on l. 174.

[Response to comments 3.19] Please see [Response to comments 3.16].

[Comments 3.20] l. 212, Fig. S13: Check SOC values, they seem small for 0 - 100 cm

[Response to comments 3.20] We plotted the SOC (0-100 cm) on a log₁₀ scale. This dataset was interpolated from measured SOC points by Rapid Carbon Assessment (RaCA) by the USDA-NRCS Soil Science Division in 2010 using Kriging method. A SOC (0-100 cm) stock map can be found at

https://www.nrcs.usda.gov/Internet/FSE_DOCUMENTS/nrcseprd1302889.pdf

[Comments 3.21] l. 216: ...overly dependent on current year's productivity. But what about the previous year's, as noted on l. 174. It might appear that DGVM Rh is too insensitive to productivity in mesic ecosystems, so that NEE varies too much with GPP.

[Response to comments 3.21] Fig 4b showed that GPP is significant positively correlated with Rh in the CONUS.

Ecosystem respiration modeling experiments

[Comments 3.22] l. 387: Include units of independent variables. I frankly place very little confidence in results from such models.

[Response to comments 3.22] We have added the units for all the independent variables. Please also see [Response to comments 3.4]

[Comments 3.23] l. 389: This equation indicates a Q₁₀ > 3, which seems large.

[Response to comments 3.23] This equation was developed by Hursh et al., (2017) by fitting models based on the soil respiration database, and has been demonstrated to be effective at simulating regional respiration (Ballantyne et al., 2017). Between 2000 and 2014, the mean annual temperature is 11.5 °C, and therefore the Q₁₀ is 2.415 (= 0.21*11.5). In fact, a recent global analysis have suggested the climatological Q₁₀ ranged from ~1 in hot climate to > 5 in cold climate (Koven et al., 2017).

Response to reviewers

1. Hursh, A. et al. The sensitivity of soil respiration to soil temperature, moisture, and carbon supply at the global scale. *Glob Chang Biol* 23, 2090-2103, doi:10.1111/gcb.13489 (2017).
2. Ballantyne, A. et al. Accelerating net terrestrial carbon uptake during the warming hiatus due to reduced respiration. *Nature Climate Change* 7, 148-152, doi:10.1038/Nclimate3204 (2017).
3. Koven, C. D., Hugelius, G., Lawrence, D. M. & Wieder, W. R. Higher climatological temperature sensitivity of soil carbon in cold than warm climates. *Nature Clim. Change* 7, 817-822, doi:10.1038/nclimate3421 (2017)

REVIEWERS' COMMENTS:

Reviewer #1 (Remarks to the Author):

Authors appropriately addressed all of my concerns on the previous manuscript, and now I can encourage publishing this manuscript in the Nature Communications.

Reviewer #2 (Remarks to the Author):

The authors have nicely provided responses and revisions addressing my previous concerns. I'm confident that this paper will now provide an interesting topic for discussion in the literature.

Reviewer #3 (Remarks to the Author):

I have no further comments on the findings in this paper, which can now proceed to publication. I would ask the authors to note in their concluding statement that the inferences drawn from this study should be confined to CONUS. There is ample evidence to suggest that GPP and thereby NEP in deciduous boreal forests to the north of CONUS respond positively to temperature in years when precipitation is non-limiting, suggesting a greater sensitivity of GPP than of ER to temperature in colder climates. The authors might also note that EC data in coniferous forests indicate that high temperatures can adversely affect GPP as well as increase ER.

Minor editorial issues:

l. 202: $p = 0.25$?

l. 215: sensitivity of microbial respiration to soil moisture increases with soil moisture?

l. 202: constraint?

Response to reviewers

Reviewer #1 (Remarks to the Author):

[Comments 1.1] Authors appropriately addressed all of my concerns on the previous manuscript, and now I can encourage publishing this manuscript in the Nature Communications.

[Response to comments 1.1] Thank you for your suggestions and comments on previous version of this manuscript. They are very helpful in guiding our revision, and greatly improve the quality of our work.

Reviewer #3 (Remarks to the Author):

[Comments 3.1] I have no further comments on the findings in this paper, which can now proceed to publication. I would ask the authors to note in their concluding statement that the inferences drawn from this study should be confined to CONUS. There is ample evidence to suggest that GPP and thereby NEP in deciduous boreal forests to the north of CONUS respond positively to temperature in years when precipitation is non-limiting, suggesting a greater sensitivity of GPP than of ER to temperature in colder climates. The authors might also note that EC data in coniferous forests indicate that high temperatures can adversely affect GPP as well as increase ER.

[Response to comments 3.1] Thank you for your suggestions and comments. We have highlighted that the conclusion drawn from this analysis is confined to CONUS in multiple places (e.g., lines 24, 92, 263).

Minor editorial issues:

[Comments 3.2] l. 202: $p = 0.25$?

[Response to comments 3.2] This is a typo. It should be $r = 0.25$ (line 204).

[Comments 3.3] l. 215: sensitivity of microbial respiration to soil moisture increases with soil moisture?

[Response to comments 3.3] A recent study (below) has shown that sensitivity of microbial respiration to soil moisture increases as soil become wetter. To avoid confusion, we have changed this sentence to read “sensitivity of microbial respiration to soil moisture increases in wetter ecosystems”. (line 217-218)

Response to reviewers

Hawkes, C. V., Waring, B. G., Rocca, J. D. & Kivlin, S. N. Historical climate controls soil respiration responses to current soil moisture. PNAS 114, 6322-6327, doi:10.1073/pnas.1620811114 (2017).

[Comments 3.4]l. 202: constraint?

[Response to comments 3.4] Suggestion adopted. (line 233)